# EverAnalyzer: A Self-Adjustable Big Data Management Platform Exploiting the Hadoop Ecosystem

**Panagiotis Karamolegkos, Argyro Mavrogiorgou** **, Athanasios Kiourtis ***** and Dimosthenis Kyriazis**

Department of Digital Systems, University of Piraeus, 18534 Piraeus, Greece
* Correspondence: kiourtis@unipi.gr; Tel.: +30-210-414-2764

**Abstract:** Big Data is a phenomenon that affects today's world, with new data being generated every second. Today's enterprises face major challenges from the increasingly diverse data, as well as from indexing, searching, and analyzing such enormous amounts of data. In this context, several frameworks and libraries for processing and analyzing Big Data exist. Among those frameworks Hadoop MapReduce, Mahout, Spark, and MLlib appear to be the most popular, although it is unclear which of them best suits and performs in various data processing and analysis scenarios. This paper proposes EverAnalyzer, a self-adjustable Big Data management platform built to fill this gap by exploiting all of these frameworks. The platform is able to collect data both in a streaming and in a batch manner, utilizing the metadata obtained from its users' processing and analytical processes applied to the collected data. Based on this metadata, the platform recommends the optimum framework for the data processing/analytical activities that the users aim to execute. To verify the platform's efficiency, numerous experiments were carried out using 30 diverse datasets related to various diseases. The results revealed that EverAnalyzer correctly suggested the optimum framework in 80% of the cases, indicating that the platform made the best selections in the majority of the experiments.

**Keywords:** Big Data; data management; data collection; data analysis; data processing; Hadoop; MapReduce; Spark; Mahout; MLlib

## 1. Introduction

Global internet consumption has increased due to the growth of the Internet of Things (IoT) and the extensive use of social media. As a result of this rise, vast amounts of data have accumulated, which in most of the cases are extremely difficult to be handled. According to Statista [1], the total amount of data consumed globally has increased to 64.2 Zettabytes in 2020, 79 Zettabytes in 2021, and is expected to increase by more than 180 Zettabytes by 2025. At the same time, Forbes [2] estimates that more than 150 Zettabytes of real-time data will be required for analysis by 2025. Companies dealing with structured data have different requirements than companies dealing with unstructured data, according to Forbes, which discovered that over 95% of organizations require assistance in managing unstructured datasets.

All this information is referred to as Big Data, which is defined as massive volumes of data collected from multiple sources and formats [3]. Many businesses gather and analyze data from various sources to make better business decisions regarding their customers, market demands, and trends. For these purposes, various Big Data processing and analysis technologies have been created to efficiently extract information from these large datasets in order to successfully evaluate the underlying data [4]. Among those tools, the ones created upon the Apache Hadoop Ecosystem are the most widely used [5]. Hadoop has become one of the most well-known tools in the Information Technology (IT) business and academic environment, due to its capacity to manage huge amounts of data.

However, as modern internet users generate massive amounts of unstructured data, the need for memory resources is increasing as well [6], with distributed data processing being a good answer to the demand for increased memory resources [7]. In this regard, two of the most widely used tools for data processing distribution are the open-source tools of MapReduce [8] and Spark [9], which provide effective solutions for processing and analyzing massive amounts of data, while providing useful functions to developers who can easily exploit them via Application Programming Interfaces (APIs) [10]. Both tools are based on the Hadoop Ecosystem, where MapReduce is used to process data in a processing cluster in parallel, whereas Spark is another solution that has been built for clustered data processing [11]. However, Spark's major purpose is to provide a programming model that can be utilized in any form of Big Data application that is constrained by the MapReduce features, while remaining error tolerant [12]. Spark is not only an alternative to MapReduce, but it also provides a variety of real-time data processing functionalities. The aforementioned tools serve as the basis for the tools of Mahout [13] and MLlib [14], which are used to perform Big Data analysis using Machine Learning (ML) algorithms [15].

The purpose of this research is to develop and deploy EverAnalyzer, a flexible Big Data management platform capable of automatically gathering, pre-processing, processing, and analyzing both real-time (i.e., streaming) and stored (i.e., batch) data. Nevertheless, most of the existing Big Data management platforms already support such a pipeline, exploiting, however, off-the-shelf technologies and tools. In addition, these platforms support tools that perform standalone tasks, such as individual data processing or individual data analysis tasks. Hence, using those platforms, specific frameworks are exploited, having their own set of benefits, shortcomings, and limitations. The solution to this problem is the implementation of a system that can comprehend the advantages and disadvantages of the various tools used to manage diverse case datasets for pursuing a processing or analytical activity and identify the optimum tool per case for performing less time-consuming and more efficient actions. EverAnalyzer comes to bridge exactly this gap, providing the innovation that enables its system to automatically recognize which of the underlying data processing (i.e., MapReduce or Spark) and data analysis (i.e., Mahout or MLlib) tools are most suitable for successfully and efficiently processing and analyzing the ingested data. The system's choice is influenced not only by the amount of data, but also by the execution speed of prior processing and analysis tasks that have been applied on relevant data scenarios. As a result, EverAnalyzer may be applied to a wide range of scenarios, better assisting users in both processing and analytical activities, hence decreasing their overall workload. To verify all of the above, the platform was evaluated through an experiment that assesses EverAnalyzer's capability to provide empirical suggestions to its users about the best framework to be utilized for the operations that they wish to perform. Data was collected from thirty (30) distinct datasets related to various diseases and conditions in the healthcare sector. The data was pre-processed, processed, and analyzed, while EverAnalyzer provided a suggestion for the most suitable framework (i.e., MapReduce or Spark for processing tasks, and Mahout or MLlib for analysis tasks, respectively) based on the shortest execution time for the requested processing/analysis process. All the framework's suggestions were gathered and compared with the framework that had the best execution time between the two chosen tools, revealing that EverAnalyzer made a correct recommendation 80% of the time. However, when the number of datasets increased, this percentage appeared to climb monotonically. This means that each performed processing/analysis task trains EverAnalyzer to export better and more representative results. Hence, if the platform uses a larger number of datasets, it is expected that the percentage of correct answers will be increased, raising the overall platform's accuracy to a percentage greater than 80%.

The remainder of this paper is organized as follows. Section 2 offers a detailed summary of the literature review that was conducted to assess meaningful insights for the study, focusing on Big Data and its lifespan, focusing in particular on the processing and analysis phases. In Section 3, a thorough analysis of how the proposed platform (EverAnalyzer) is designed and built is presented, including the platform's goals and

users as well as its architecture. Section 4 depicts the experimentation results generated by EverAnalyzer, and Section 5 provides an interpretation of the exported results as well as how they can be interpreted in relation to the studied literature. Finally, Section 6 contains the study's conclusions, limitations, next steps, and future research directions; it also describes future experiments that would be interesting to conduct using EverAnalyzer's design and implementation guidelines.

## 2. Literature Review

Big Data is defined as large volumes of data collected from various sources and in various formats [3]. Such data have some specific characteristics (Vs of the data), which primarily refer to data Volume (i.e., data size), Variety (i.e., data format), Velocity (i.e., data production rate), Veracity (i.e., size of data authenticity), Validity (i.e., data validity), Volatility (i.e., time of data validation), and Value (i.e., data usefulness in terms of analysis) [3]. These characteristics indicate that Big Data is challenging to be managed, but when it is properly managed, it may be highly valuable. For this purpose, companies can use Big Data to evaluate and extract important information about their products and customers. However, due to the wide range of their forms and sizes, analyzing them is sometimes a complicated and time-consuming task. At the same time, people are increasingly using the Internet to help them with their everyday activities and entertainment, which causes the amount of collected data to increase year after year.

This results in data that may be structured, semi-structured, or even unstructured, making them difficult to manage with traditional Relational Database Management Systems (RDBMS), which are expensive and time-consuming to implement [16]. Structured data refers to data that are known for the information they contain and the manner in which they are contained. Semi-structured data, on the other hand, lacks some specifications about the information they contain, whereas unstructured data conveys no information on their structure. Large amounts of these data can be collected by mobile phones, sensors, Global Positioning System (GPS) signals, social media, and other sources that generate massive amounts of data every second [17]. As a result, Big Data refers to either batch data deriving from ready-to-use datasets that require some processing or analytic activities (e.g., already stored data derived from external systems' databases), or streaming data derived from live sources that are constantly streaming information (e.g., real-time data gathered from social media) [18].

As a result, managing Big Data throughout their lifecycle has become a very challenging task that never ceases to pique the interest of enterprises and researchers. More specifically, the utilization of Big Data is represented by a lifecycle that includes a plethora of phases, beginning with collection of the data and concluding with their final destruction [19]. Figure 1 depicts all of these phases, referring to the: (i) collection, in which data are collected from various sources, most of the time in formats that are difficult to handle due to their unstructured nature; (ii) storage, in which the ingested data are stored in the appropriate database; (iii) processing, in which data are pre-processed in a standard structure to make it easier to manage in subsequent phases; (iv) analysis, in which various ML methods are used to produce meaningful results and insights from the stored data; (v) utilization, in which the extracted results and gained insights are put to use in a variety of real-life and testing scenarios; (vi) destruction, the final and most important phase of the entire lifecycle, since many sensitive data may be collected from various sources during the collection phase, requiring the data's compliance to a strict protocol to ensure that their confidentiality, integrity, and availability are not compromised. To this end, it should be emphasized that the suggested platform's purpose is to investigate the phases of collection, storage, processing, and analysis, which are further analyzed below.

**Figure 1.** Big Data lifecycle.

*2.1. Big Data Collection*

Big Data collection is described as the process of gathering massive amounts of data in order to further analyze them and obtain useful results [20,21]. These data can be collected using traditional methods such as questionnaires and interviews; however, there is a plethora of more effective approaches. Web services, sensor-equipped devices such as mobile phones and tablets, and smart transportation cards, are just a few examples [22]. All the data collected from these devices may be either batch, meaning that they are collected up to a predefined size and then stored all together to be analyzed later as a set of data, or streaming, referring to data that are analyzed while being collected. The distinction between those two kinds of data is that streaming data processing is applied directly to the ingested data, whereas batch data processing collects and preprocesses a predetermined quantity of data [18]. Furthermore, if it is not possible to collect enough data for a processing/analytical activity, there are methods for creating synthetic data [23], which represent the real data that an analysis would most likely use to properly execute the required analysis.

Various tools, such as Sebek [24], Hflow [25], Honeywall [26], Nepenthes [27], Kojoney [28], and Capture-HPC [29] have been built to successfully collect such varied types and formats of data. Kafka [30] and Flume [31] are two of the most widely used data collection tools. Whereas Kafka is a streaming data collection and processing tool, Flume is primarily used to manage infrastructures for collecting streaming data as batch data. Flafka is created by combining those two tools, providing the ability to save streaming data as batch data exploiting both Kafka and Flume [32].

*2.2. Big Data Storage*

Big Data storage is described as the process of storing and managing large-scale datasets while maintaining data access reliability and availability [33,34]. Big Data storage has a significant impact on the infrastructure of the system that desires to adopt it. On the one hand, the storage infrastructure must provide reliable space to storage services, but on the other hand, it must also provide a dynamic access interface for querying and analyzing large amounts of data.

Because the volume of Big Data is continuously expanding, complex systems known as Database Management Systems (DBMS) are increasingly being employed to store and manage these data. Structured Query Language (SQL) systems and Non-SQL (NoSQL) systems are the two representative types of RDBSs [35]. NoSQL systems are preferable for storing and managing Big Data, since SQL systems require organized data to be efficient, whilst NoSQL systems are meant to be used for unstructured data. To better manage the variety of the forms of the existing unstructured data, NoSQL DBMSs are classified into three separate core categories, namely: (i) key-value stores that store data as a collection of key-value pairs in which a key serves as a unique identifier, with both keys and values ranging from simple objects to complex compound objects (e.g., Redis [36]; Scalaris [37], Tokyo Tyrant [38], Riak [39]); (ii) document stores that are databases for storing information in the form of documents (e.g., SimpleDB [40], CouchDB [41], MongoDB [42], Terrastore [43]); (iii) column stores that use tables, rows, and columns, but unlike a relational database, the names and format of the columns can vary from row to row in the same table (e.g., Bigtable [44], HBase [45], HyperTable [46], Cassandra [47]).

*2.3. Big Data Processing*

Big Data processing is a group of techniques for accessing large amounts of data in order to extract meaningful information for decision support and provision [48,49]. Big

Data processing employs a range of methods, such as wordcount and string matching, which can be distributed across vast clusters of processing units [50]. Data processing algorithms typically have low algorithmic complexity, allowing them to perform quick computations. They are simple to implement and can interpret a variety of datasets, whereas they may be used on any dataset, regardless of its size, due to their high speed. However, directly obtained datasets (i.e., raw datasets) are frequently impossible to process as a data processing task, since in the case of Big Data such datasets do not comply with a specific structure as they derive from a broad range of sources. Thus, Big Data must first go through a data pre-processing phase to normalize the data structure before going through a data processing job. After the data structure is normalized, it is then simple to process the data using the preferred data processing algorithms.

At the same time, traditional programming paradigms are incapable of handling data effectively because it is often stored on thousands of commodity servers. As a result, new parallel programming methods are being deployed in datacenters to improve the performance of NoSQL databases [48]. MapReduce is a popular programming model for Big Data processing on large-scale commodity clusters, and it has evolved as an important component of the Hadoop ecosystem [48]. The main advantage of this programming model is its simplicity, which allows its users to easily exploit it for Big Data processing tasks [51]. Pig is an SQL-like environment that is used for performing processing tasks upon Big Data [52], whereas Hive is another example of such tool that provides a better environment than MapReduce and simplifies the code development as programmers are not required to deal with the complexities of MapReduce coding [53]. Similarly, many solutions have been developed to address MapReduce's gaps, such as delayed data loading and data reuse. Among those tools are Starfish, which is a Hadoop-based framework aiming to improve the performance of MapReduce jobs through the use of data lifecycle analytics, as well as being a self-tuning system that adapts to users' needs and systems' workloads without requiring users to configure or change the underlying settings or parameters [54]. Spark is an alternative to MapReduce that aims to overcome disk I/O limitations and improve the performance of prior solutions. The ability to perform in-memory computations is the main feature that distinguishes Spark, since it enables data to be cached in memory, removing the disk overhead limitation of MapReduce for iterative tasks [55]. Other programming models similar to MapReduce include Dryad, which is a distributed execution engine for running Directed Acyclic Graph-based (DAG) Big Data applications. While MapReduce only allows for a single set of input and output data, Dryad allows users to use any number of input and output data [56]. Pregel is another tool capable of processing large-scale graphs for a variety of purposes, including network graph analysis and social networking services [57]. Finally, data processing technologies are available for streaming data as well. As data is acquired from their source, these technologies provide processing workflows, removing the requirement to convert data to batch data [58]. Examples of such tools are Storm [59], Flink [60], Spark Streaming [61], Samza [62], Apex [63], and Google Cloud Dataflow [64], among others.

*2.4. Big Data Analysis*

Big Data analysis is defined as the procedure for acquiring data from diverse sources, processing them to extract relevant patterns and insights, and distributing the results to the appropriate stakeholders [65,66]. Data analysis is classified into four (4) discrete types, which refer to: (i) descriptive analytics that respond to the question "What happened?" and mines information from raw data; (ii) diagnostic analytics that report on the past while attempting to answer the question "Why did it happen?"; (iii) predictive analytics that answer future-related questions "What will happen?" and "Why will it happen?"; (iv) prescriptive analytics that suggest one or more courses of action and illustrate the likely outcome/influence of each action, providing answers to the questions "What should I do?" and "Why should I do it?", based entirely on "what-if" scenarios. In this context, there have been numerous techniques and methodologies proposed for addressing each

analysis question [67,68]. Some of the most indicative techniques refer to: (i) clustering, which is used to explore data and finding natural groupings; (ii) classification, which is used for predicting a specific outcome; (iii) association, which finds rules associated with frequently occurring items; (iv) regression, which is used for predicting a continuous numerical outcome; (v) attribute importance, which ranks attributes based on the strength of their relationships with target attributes; (vi) anomaly detection, which identifies cases that are unusual or suspicious based on deviations from the norm; (vii) feature extraction that creates new attributes by combining the existing ones in a linear fashion.

Numerous data analysis tools, largely open-source, have been developed to implement such analytical approaches. Orange is a data analysis and visualization tool that includes components for ML feature selection and text mining [69]. R is a statistical computing and graphics programming language and software environment that is widely used for the development of statistical software and data analysis [70]. Weka is a data mining software that offers a set of ML algorithms for data mining activities, such as classification, regression, clustering, association rules, and visualization [71]. Moreover, Shogun is a software toolbox that provides a wide variety of algorithms and data structures for ML problems, with a concentration on data mining tasks, including regression and classification [72]. Rapid Miner is another data manipulation, analysis, and modeling tool that operates through visual programming [73]. In addition to open-source tools, there are various commercial data mining solutions like Neural Designer [74], SharePoint [75], Cognos [76], and BOARD [77]. While the technologies listed above are intended for performing and visualizing analytics, they do not follow the Big Data logic. Tools built in this manner are Sisense, a business intelligence platform that can join, analyze, and visualize data [78], KEEL, which assists users in evaluating evolutionary algorithms for data mining problems [79], Mahout, which provides clustering, classification, and batch-based collaborative filtering algorithms that run on top of MapReduce [13], and MLlib, a project with a similar approach to Mahout, which executes its ML algorithms on top of the Spark framework to make extensive use of the system's Random Access Memory (RAM) [80].

## 3. Proposed Big Data Management Platform

Following the established baseline concepts, this section describes the architecture of the EverAnalyzer platform, which has been built to cover the entire Big Data lifecycle towards the successful collection, storage, processing, and analysis of either streaming or batch data. The platform has been built around the Hadoop ecosystem and its various supported frameworks, specifically exploiting the Kafka and Flume frameworks for collecting the underlying data, MongoDB and Hadoop Distributed File System (HDFS) for storing all the gathered data, the Spark and MapReduce frameworks for effectively processing this data, as well as the ML libraries of MLlib and Mahout for finally analyzing the data and extracting useful insights. All these tools have been selected based on the thorough literature review that has been conducted (further examined in Section 5), which indicated that these tools constitute the most popular and widely used Big Data collection, storage, processing, and analysis frameworks, being rapidly studied and applied by both industry and academia. Based on the research that has been conducted, even though Kafka, Flume, MongoDB and HDFS constitute a common path for data collection and storage in a wide range of research works, what is of major importance are the frameworks that should be put in place to efficiently manage the data processing and analysis paths. Towards this goal, there appears not to have been any attempt to manage data processing and analysis using a hybrid system that could apply the most efficient of the processing frameworks of Spark and MapReduce and the analysis frameworks of MLlib and Mahout to the corresponding Spark and MapReduce framework. The platform of EverAnalyzer tries to bridge this gap, not only by providing a solution to the users for managing their processes using Spark or MapReduce, but also by allowing the platform to advise users on the best framework for more efficiently and quickly processing and analyzing their

collected data. With this feature, the platform will be a valuable resource for any user looking to exploit the Hadoop ecosystem.

In deeper detail, EverAnalyzer's objectives include the creation of a platform that includes a registration subsystem that allows users to login and logout of their accounts. This registration system will ensure the platform's users' confidentiality, integrity, and availability. Aside from the registration subsystem, EverAnalyzer offers a Data Collection subsystem that allows its users either to upload their own batch data into the platform, or to collect streaming data from the Twitter database [81]. Additionally, a data processing and analysis subsystem is provided, allowing users to perform configurable pre-processing, processing, and analytics on their collected data, which is finally stored in the platform for future use or access. Within these configurations, users can use Spark or MapReduce to process their data, as well as analyze them using their respective MLlib or Mahout libraries. The platform can respond to the optimal use of Spark or MapReduce and the appropriate MLlib or Mahout libraries, depending on the tasks that the users wish to complete. Finally, the results of the users' analytics are graphically displayed in a user-friendly manner and can be exported for future use. All these objectives are summarized in Table 1.

**Table 1.** EverAnalyzer objectives.

| ID | Objective |
|---|---|
| #1 | Users can login to the platform with a registration system. |
| #2 | Users can collect their desired batch or streaming data. |
| #3 | Users can save their collected data and the analyzed data for future use or access. |
| #4 | Users have the ability to pre-process their data. |
| #5 | Users are recommended by the most suitable processing framework to be applied, where the platform recommends the optimal use between Spark or MapReduce. |
| #6 | Users are recommended by the most suitable analysis framework to be applied, where the platform suggests the optimal use between MLlib or Mahout libraries. |
| #7 | The users' analytics results are displayed graphically, in a user-friendly way. |

*3.1. Platform Architecture*

Figure 2 shows a high-level overview of the platform's architecture. As shown in the figure, EverAnalyzer contains various subsystems that deal with user interaction as well as data management and storage within the platform. These subsystems create the entire platform and work together to achieve the platform's objectives depicted in Table 1.

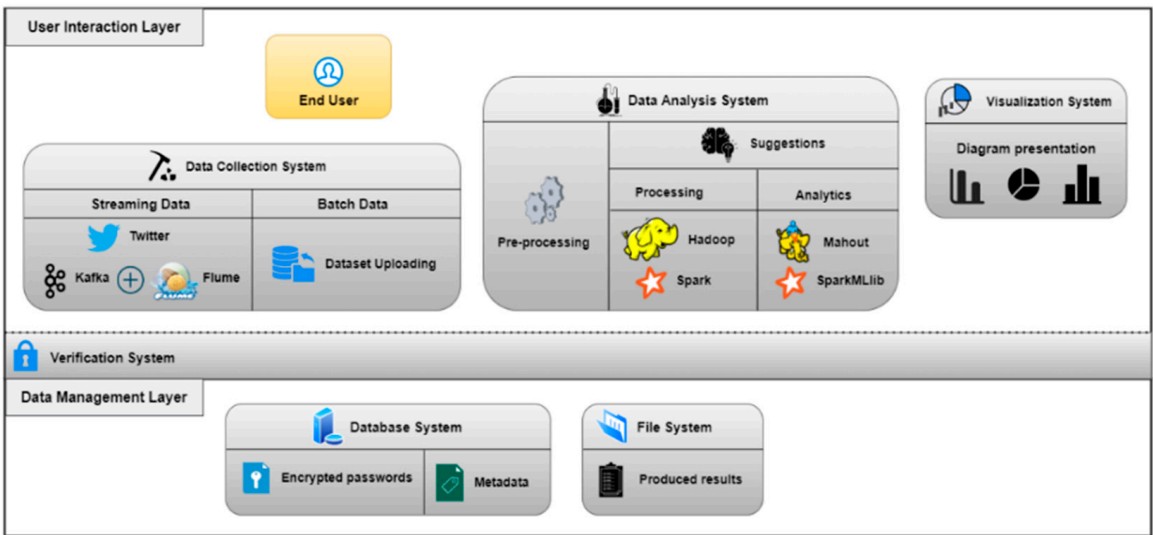

**Figure 2.** EverAnalyzer High Level Architecture.

More specifically, the provided architecture is divided into two layers, the *Data Management Layer* and the *User Interaction Layer*. The *User Interaction Layer* contains the subsystems with which the users directly interact. In contrast to the preceding, the *Data Management Layer* is primarily managed by the platform's system. As a result, the users interact with the containing subsystems indirectly through the data that they provide in the *User Interaction Layer*. To be more specific, the *Data Management Layer* and the *User Interaction Layer* subsystems include the:

- Verification System: Users can register and login to the platform.
- Data Collection System: Users are able to provide the configurations required to collect the desired data.
- Data Analysis System: Users can configure the analytic jobs they want to run and then execute them to acquire their results.
- Visualization System: Users can receive visual results of their analytic jobs/processes.
- Database System: This is only used by the platform and not the users since it saves useful data for each process and each platform's user.
- File System: This is only used by the platform and not by the users, as it distributes across multiple machines all the users' analyzed, pre-processed, processed, and collected data.

Table 2 depicts the subsystems' association with the platform's overall objectives.

**Table 2.** Correspondence of subsystems and EverAnalyzer objectives.

| EverAnalyzer Layer | Subsystem | Objective |
|---|---|---|
| *User Interaction Layer* | Verification System | #1 |
| | Data Collection System | #2 |
| | Data Analysis System | #4, #5, #6 |
| | Visualization System | #7 |
| *Data Management Layer* | Database System | #1, #3 |
| | File System | #3 |

Figure 3 displays a low-level representation of the platform's architecture, indicating how all of the platform's subsystems interact.

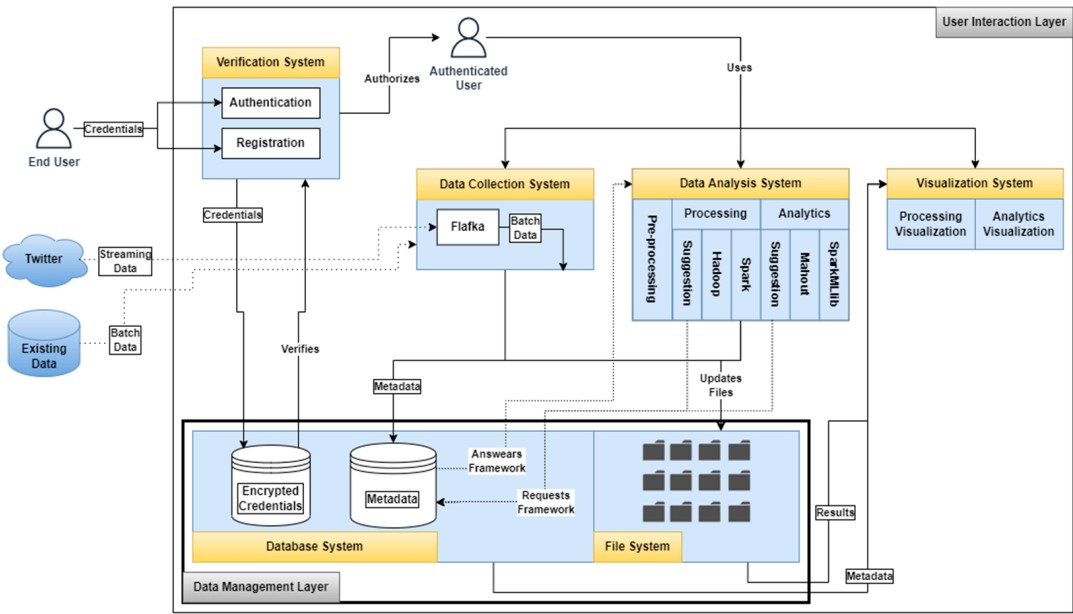

**Figure 3.** EverAnalyzer Low Level Architecture.

In more detail, initially the user can use the Verification System to authenticate and register onto the platform. This system then communicates with the Database System to determine whether the user can enter the platform or not. After the user is authenticated, they can access the Data Collection System, the Data Analysis System, as well as the Visualization System. The Data Collection System demonstrates the use of Flafka, a combination of Flume and Kafka to obtain the requested data (either in a batch form or in a streaming form by Twitter). Afterwards, the Data Analysis System allows the user to select an action among pre-processing, processing, and analytics tasks, as well as to obtain a recommendation for the framework that would better fit and perform in the user's data scenario. This proposal is provided once the Data Analysis System receives the response from the Database System. Following the use of the Data Collection and Data Analysis Systems, new metadata is provided to the Database System, while the appropriate updates are made to the File System's files. Finally, the Visualization System pulls the metadata and the results from all the other systems via the Data Management Layer in order to perform the visualizations requested by the user. All the above-mentioned functionalities of all the subsystems are thoroughly explained in the following sections.

### 3.1.1. Verification System

The platform's Verification System exists to authenticate different users, supporting the features of: (i) user login and authentication; (ii) new user registration in the system; (iii) user logout of the system; (iv) check of existing or non-existing users within the system from the browser session where the platform is displayed, to prevent non-authenticated users from abusing the platform.

### 3.1.2. Data Collection System

The platform's Data Collection System allows its users to collect either batch data or streaming data from the Twitter database. For streaming data, the following features are supported: (i) customization of the number of tweets to be collected by the user; (ii) customization of the words to be searched in the Twitter database by the user; (iii) naming of user's collected dataset; (iv) gathering of data based on user's configurations directly from Twitter's database. For batch data, the following features are supported: (i) finding the location of user's filesystem for the data storing; (ii) uploading the dataset as batch data; (iii) customizing the words searched in the uploaded dataset; (iv) naming the user's uploaded dataset.

### 3.1.3. Data Analysis System

The platform's Data Analysis System is the most complicated and the most critical component of the platform. In essence it is the subsystem that allows the users to pre-process, process, and analyze their datasets. Its features are divided into three categories. All the categories with their supported features are listed below:

Pre-processing: (i) The platform's system determines whether data has been collected by the user so that it can be pre-processed or not. If no datasets are collected, the system notifies the user, whereas if data is gathered, the system displays to the user all the datasets that it has collected, displaying useful metadata about them, such as labels, words used for the collection and data size. (ii) After selecting the collected dataset for pre-processing, users specify which data/tweets they want to keep based on their contained fields. (iii) After selecting the collected dataset for pre-processing, users can specify which fields/attributes should be retained in the new pre-processed dataset.

Processing: (i) The platform's system determines whether pre-processed data exists or not, so that the user can proceed with a word count processing phase. If no pre-processed datasets exist, it notifies the user of their need to perform the desired processing job; otherwise, the system displays relevant metadata about the datasets, such as labels, words used for the collection, data size, and performed pre-processing results. (ii) After selecting the pre-processed data for processing, the user is given the option of executing a word count

process using MapReduce or Spark. (iii) If the user has not decided which framework to use, the platform recommends to the user the most appropriate one. This proposal is based on the system storage and the available RAM of the machine that the platform is installed on. In addition to the aforementioned, the platform's system examines prior processes, if any, to provide an answer based on the knowledge gathered by the system. This feature is considered of great importance for the platform, since the main goal of its developed system is to be able to apply the most efficient and least time-consuming framework between the processing frameworks of Spark and MapReduce. According to the literature, most of the time, Spark is faster than MapReduce on the same use cases, using the same resources. As the volume of datasets grows in different use cases, MapReduce is expected to outperform Spark in terms of execution speed, but only until the distributed cluster's RAM is insufficient for Spark. When the dataset is small enough to fit in the system's RAM, Spark is the most suitable choice for processing jobs. EverAnalyzer utilizes this fact to guess the better solution, after which MapReduce will be superior for the users. In more detail, the platform's system saves the execution speed of the process and the framework used for each completed process, following the flow depicted in Figure 4, which is triggered for each proposition query. Figure 4a indicates the flow followed for the MapReduce proposition, where the users get the current dataset, and the system finds all the processing jobs that used MapReduce for their processing and datasets smaller than the current one. The system starts iterating the MapReduce processing jobs, and for each iterated job it tries to find any completed Spark processing job that used a dataset even smaller than the current processing job is using. If it finds any Spark processing job that was slower than one of the iterated MapReduce processing jobs, then the system proposes MapReduce. On the other hand, Figure 4b illustrates that if there has not been any suggestion yet by the system, then it finds all the processing jobs that used MapReduce for their processing and had larger datasets than the current one. The system starts iterating the MapReduce processing jobs, and for each iterated processing job it tries to find any completed Spark processing jobs that used a dataset even larger than the one currently being used. If it finds any Spark processing that is faster than any of the iterated MapReduce processing jobs, then the system proposes Spark. If there is no proposition yet, then the system randomly chooses one of the frameworks, with a 50% chance for each framework being chosen.

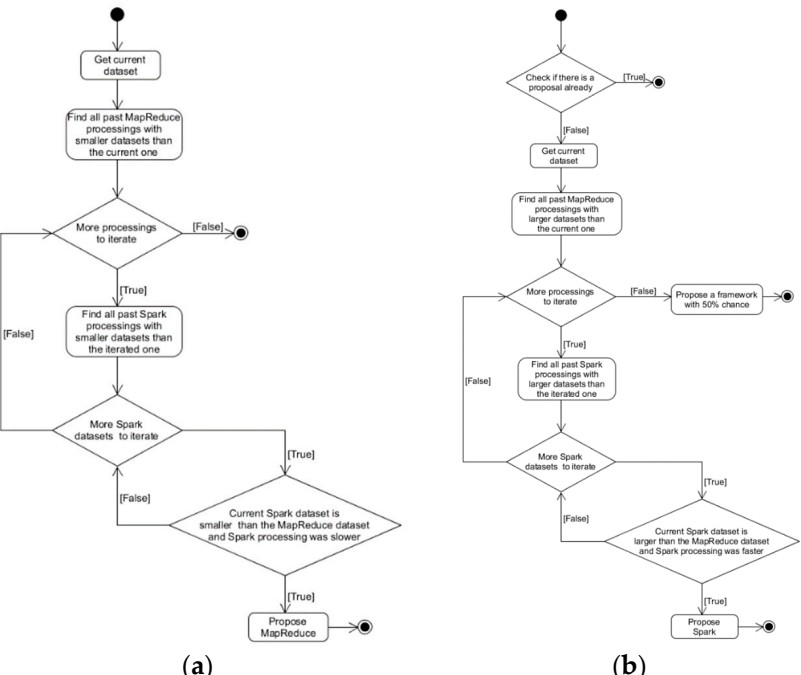

(**a**)  (**b**)

**Figure 4.** (**a**) MapReduce proposition flow; (**b**) Spark proposition flow.

Analytics: (i) The platform's system determines whether there are pre-processed data that the user can use to create an analytic job, and if not, it informs the user of their need to perform this process. If there are pre-processed data, the platform displays useful metadata about the data, such as labels, words used for the collection, data size, and performed pre-processing results. (ii) After selecting the pre-processed dataset, the user can perform an analysis using the Mahout or MLlib libraries. (iii) If the user has not determined which framework to use, the platform suggests one. This proposal is based on the amount of available RAM and the dataset that the user wishes to analyze. This feature is also considered highly important to the platform. Unlike the processing proposition flow (Figure 4), the analytics proposition flow is monotonous. According to the literature, MLlib will always outperform Mahout in terms of execution speed, except when the dataset for analysis is larger than the RAM of the platform's system. MLlib has been shown to crash when it is unable to handle entire datasets within the system RAM, which has been shown to be much slower in these cases. As a result, the algorithm used for the framework proposition to be used for the analytics proposition flow only considers the system's RAM. If the system's RAM is less than the size of the given dataset, the proposal is Mahout. If the system's RAM is larger than the size of the given dataset, the proposal is MLlib. Figure 5 depicts the concept of this proposition flow.

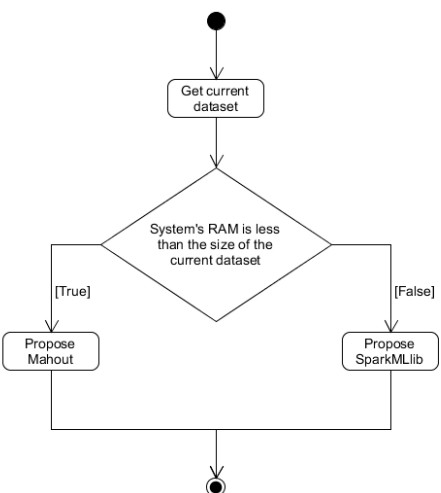

**Figure 5.** Analytics proposition flow.

### 3.1.4. Visualization System

The platform's Visualization System is used to display to the user the extracted results of each processing and analytic job. The purpose of this action is to aid the user in understanding the produced data analysis results. The visualization can be explained in two (2) steps: (i) The platform determines whether there are processed or analyzed data that the user can visualize or not. If no such data exist, the platform's system notifies the user that the visualization requires processed or analyzed data, or else the analyzed data appear alongside various metadata related to the user's analysis, such as data labels, words used for the data collection, data size, pre-processing performed, and analysis applied. (ii) When the user selects the analyzed data to be visualized, the platform displays the corresponding visualizations.

### 3.1.5. Database System & File System

The Data Management Layer, which includes the platform's Database System, and File System interact with all the platform's subsystems, being responsible for holding all the exported data from each user and each implemented activity. In addition to the foregoing, this layer's subsystems give the user access to all the generated and collected data.

*3.2. Platform Users*

EverAnalyzer was created and built to be used by a wide range of users, as illustrated in the use case diagram in Figure 6. Some indicative examples of such users may include: (i) Data Analysts who may use the platform to extract and interpret results from their data; (ii) data scientists who may be able to experiment with different methods and tools for extracting results from their collected data; (iii) data engineers who will be able to use all of the platform's automated processes as a tool to supplement the rest of their tools.

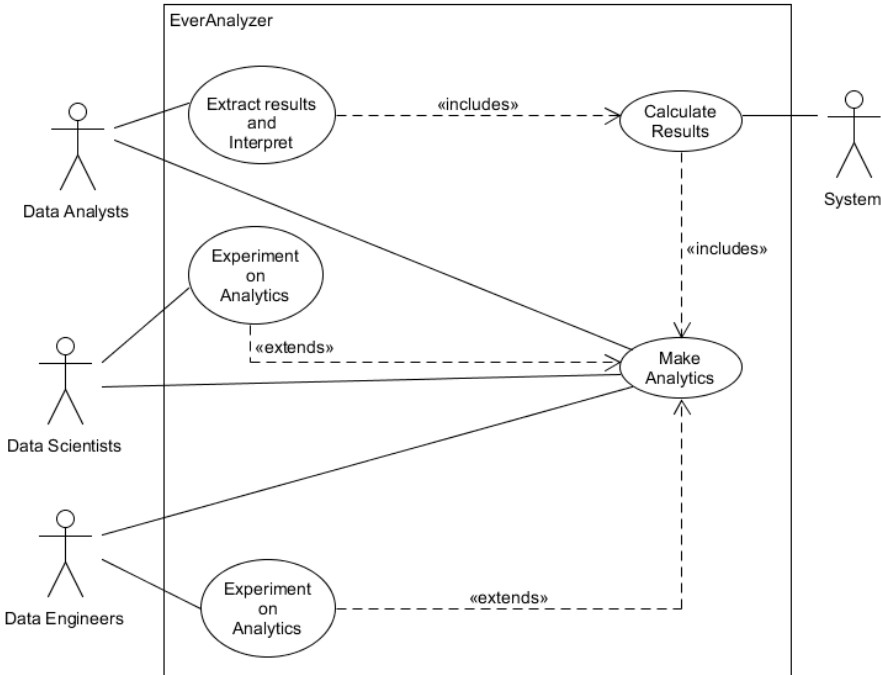

**Figure 6.** Use Case diagram of EverAnalyzer users.

Figure 7 depicts the user's interaction with the platform's system during the authentication process (i.e., Objective #1 of the platform), in which the user can register in the system with the option to login later, while the system saves and authenticates the user's data.

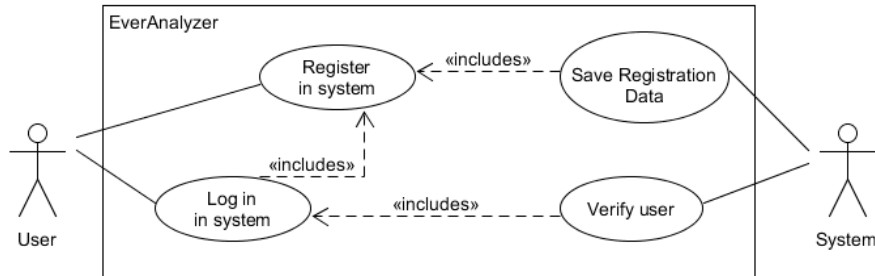

**Figure 7.** Use Case diagram of EverAnalyzer Objective #1.

Figure 8 depicts the system and user operations during the data collection process (i.e., Objective #2 of the platform), where the user can configure the data collection, while the system performs the collection and saves the data for later use.

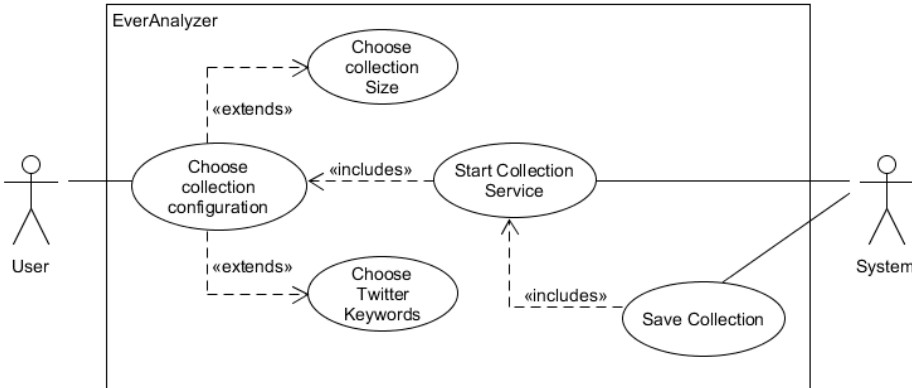

**Figure 8.** Use Case diagram of EverAnalyzer Objective #2.

Figure 9 depicts the platform's data storage process (i.e., Objective #3 of the platform), in which the user completes any data collection, pre-processing, processing, or analytic tasks, and then, as the responsible executor of the saving process, the system stores all the information about the user's data as well as the data itself in their respective storage resources.

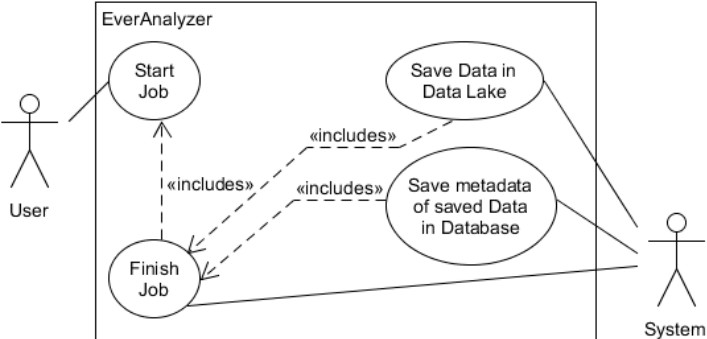

**Figure 9.** Use Case diagram of EverAnalyzer Objective #3.

Figure 10 depicts the platform's data pre-processing process (i.e., Objective #4 of the platform), in which the user can configure a pre-processing job and then the system handles the user's pre-processing task automatically, performing any configurations that are given to it.

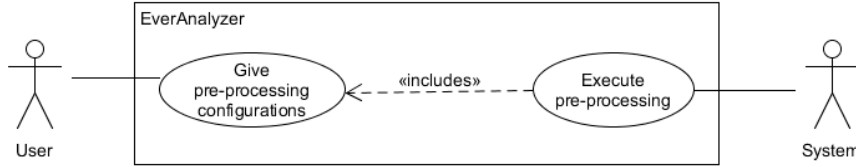

**Figure 10.** Use Case diagram of EverAnalyzer Objective #4.

Figure 11 displays the framework suggestion process that will be used in the user's processing and analytic activities (i.e., Objectives #5 and #6 of the platform), in which the user seeks a proposal from the system and updates the system with the data for the desired processing/analysis task. The system then collects the user's processing/analysis data and, after processing it, it informs the user of the best framework to select.

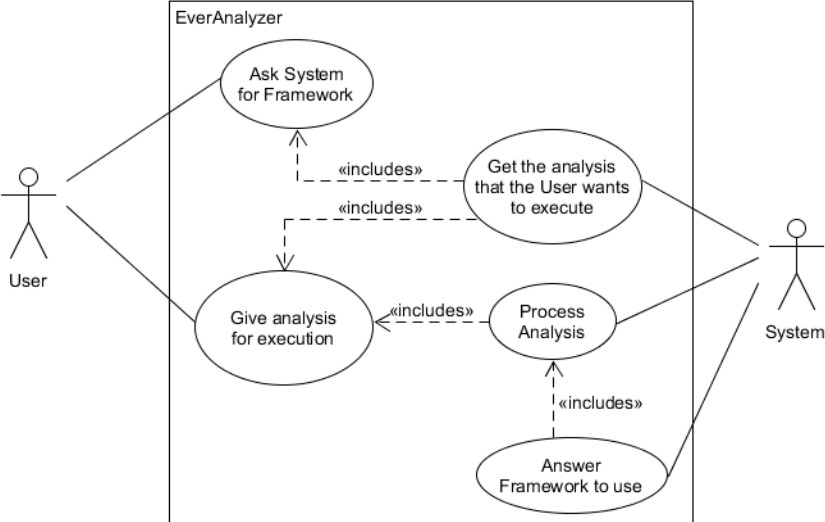

**Figure 11.** Use Case diagram of EverAnalyzer Objectives #5 and #6.

Finally, Figure 12 illustrates how the system presents the extracted results (i.e., platform objective #7), where the user picks the results for visualization while also having the opportunity to inspect and display the final images of the finished operations. The system then takes the user's preference for how the outcomes should be represented as an input and generates the appropriate diagrams of the executed procedure (i.e., processing or analysis task).

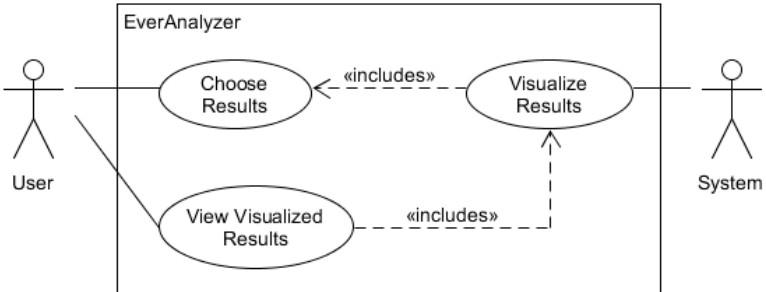

**Figure 12.** Use Case diagram of EverAnalyzer Objective #7.

## 4. Case Example

### 4.1. Working Environment

The experimentation was carried out on a system with 237 gigabytes of Hard Disk Drive (HDD) and eight gigabytes of RAM. There was no distribution because there was only one cluster node using the platform's distribution. The platform was running on a machine using Windows 10 Home. Apache Hadoop v3.1.0 (including MapReduce), Apache Flume v1.9.0, Apache Kafka v2.13-2.7.0, Apache Mahout v2.13-2.7.0, Apache Spark v3.1.2 (including MLlib), Apache Tomcat Server v9.0, and Twitter API v1.1 were the tools that were used. On top, the platform was developed using the technologies of HTML (version 5), CSS (version 3), Bootstrap (version 5) and JavaScript (version ES6) for its front-end part, and Java (version 8) for its back-end part.

### 4.2. Use Case Description

Experiments were conducted on the EverAnalyzer platform to assess its functionality and ability to offer appropriate recommendations based on the frameworks used. Specifically, each platform interface was used to create experimental processing and analytical procedures using the proposal mechanisms of the MapReduce and Spark frameworks.

Despite the fact that they supported libraries for performing the data analysis tasks (i.e., Mahout and MLlib), the experiment was solely focused on the MapReduce and Spark proposals, using the metadata collected after each user processing task to suggest the most suitable framework to be used in the investigated scenario. More precisely, the platform's functionalities were tested using various streaming health data deriving from Twitter. Thirty different diseases and health conditions were chosen, and 500 Tweets were collected as streaming data by EverAnalyzer, which were then placed within the EverAnalyzer Data Management Layer as batch data using Flafka. The experimentation used keywords from the World Health Organization's (WHO) website to describe each selected disease and condition for the collection [82]. Table 3 lists the diseases and conditions.

**Table 3.** Chosen diseases and conditions for collected data.

| Anaemia | Cancer | Cholera | Coronavirus | Influenza | Monkeypox |
|---------|--------|---------|-------------|-----------|-----------|
| Obesity | Pneumonia | Smallpox | Syphilis | Tetanus | Yellow fever |
| Zika virus | Trachoma | Diabetes | Diarrhoea | Ebola virus | Epilepsy |
| Hepatitis | HIV-AIDS | Depression | Disability | Cardiovascular | Chagas |
| Dementia | Dracunculiasis | Echinococcosis | Foodborne | Hypertension | Infertility |

### 4.3. Platform Evaluation

The platform was evaluated in two ways. The complete platform's functionality and given User Interfaces (UI) were first tested by its users, as stated in Section 4.3.1; then its proposal performance was measured, as shown in the experimentation results in Section 4.3.2. The results of the performance were further visualized and explained.

### 4.3.1. Functional Evaluation

This Section describes how EverAnalyzer is used, where each UI of the platform is presented, referring to the: (i) authentication Interface that reflects the user's registration and authorization phase; (ii) homepage Interface that depicts the EverAnalyzer's homepage; (iii) collection interface that reflects the UI, through which the platform guides the user in gathering data from the Twitter API; (iv) pre-processing interface that allows the user to pre-process the ingested datasets; (v) processing interface that allows the user to perform word count jobs on the pre-processed datasets; (vi) analytics interface that assists the user in performing analytic tasks on the pre-processed datasets; (vii) visualization interface that allows the user to see all of the datasets produced by the processing and analytics activities; (viii) management interface that contains all of the user's datasets, allowing the parsing and downloading of all the user's information.

In deeper detail, the authentication interface was created for user access in the EverAnalyzer platform. The verification system is deployed alongside it, securing the information of all the EverAnalyzer users. Figure 13a,b depict the sign-in and sign-up interfaces, respectively. More specifically, the user can use the sign-up page to provide a password and a username for use on the platform's sign-in page. The user's password is encrypted before it is saved in the database system, whereas in both interfaces the platform issues user-friendly warnings if the user performs an incorrect action, such as entering different values in the "password" and "verify password" fields or attempting to sign up with an already existing username.

After logging in to the platform, a side bar is displayed in the Homepage Interface, allowing the user to access and use the EverAnalyzer supported functionalities (Figure 14). These functionalities refer to collection, pre-processing, processing, analytics, visualization, and management, which will be covered in greater depth below.

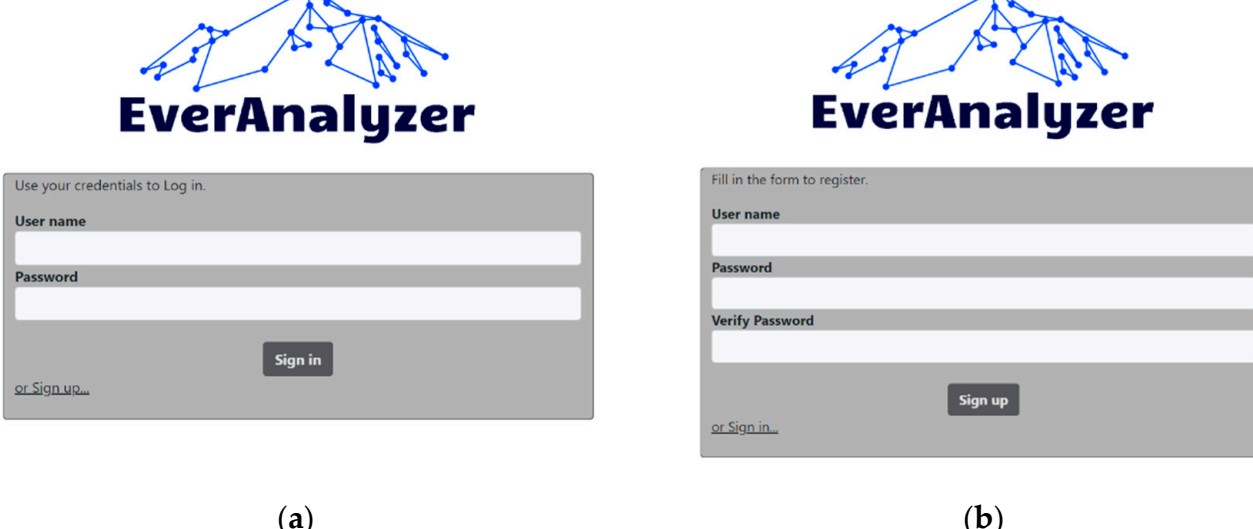

**Figure 13.** (**a**) Sign-in interface; (**b**) Sign-up interface.

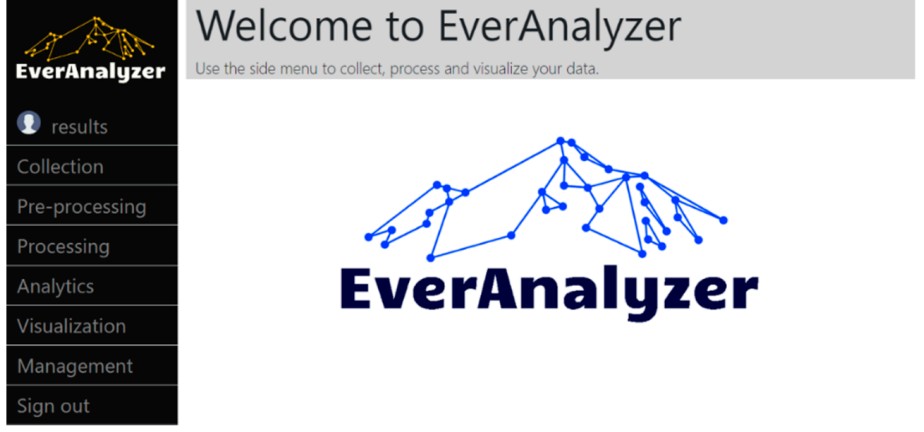

**Figure 14.** Homepage Interface.

In the collection interface, users can collect datasets from the Twitter API. They can specify the collection's label, as well as the number of Tweets to be collected and the keywords to be used. The keywords are especially necessary for collection of the data, as the Twitter API returns only Tweets that contain the keywords that the user provided. The collection interface is depicted in Figure 15. To this end, it should be noted that the UI assists the users in avoiding mistakes such as typing a label that already exists in one of their current dataset labels or failing to fill out a field. This occurs through the use of user-friendly warnings when the user performs an incorrect action, also preventing the user from submitting the collection without a valid collection form.

As shown in Figure 16a, the pre-processing interface offers all the user's collection datasets as pre-processing options, whereas a metadata summary for each collection dataset is displayed in greater detail. This metadata includes the dataset's label, the date it was collected, the number of Tweets it contains, its size in bytes, and the words used to retrieve the dataset from Twitter. When the user decides which collection dataset to pre-process, the platform displays the pre-processing form by pressing the "Select Dataset" button. The user can select which fields of the raw Tweets to keep in this form. Because of the platform's ability to detect all the fields that the user requests, regardless of how nested they are within the JavaScript Object Notation's (JSON) schema, this choice is not limited by the JSON schema. Following the pre-processing, only the Tweets with the required

fields remain in the dataset. The pre-processing form is depicted in Figure 16b. To this end, it should be noted that the UI always assists the user in avoiding mistakes such as typing a label that already exists in one of the current dataset labels or failing to fill out a field. This occurs through the use of user-friendly warnings when the user performs an incorrect action, preventing the user from submitting the pre-processing without a valid pre-processing form. Moreover, if there are no collection datasets yet, the interface notifies the user that a pre-processing job is required.

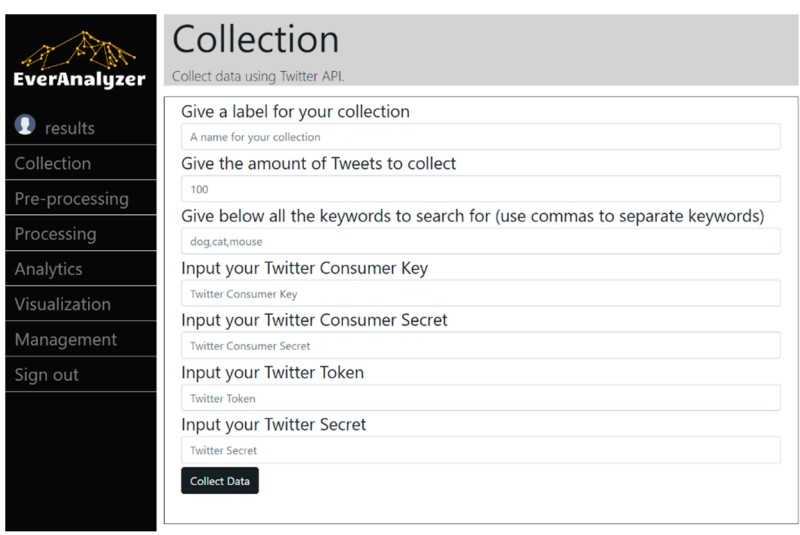

**Figure 15.** Collection Interface.

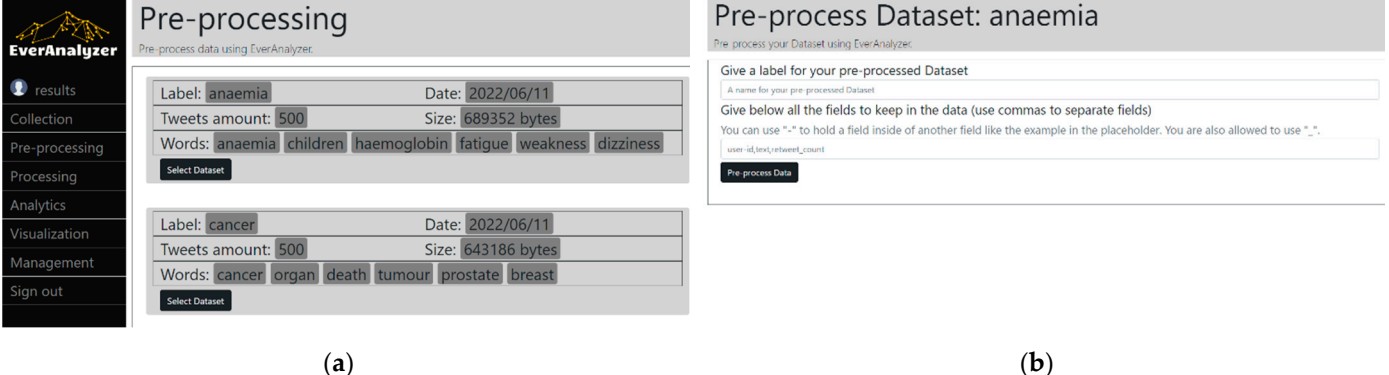

(**a**)                                                                                                                                   (**b**)

**Figure 16.** (**a**) Collected datasets; (**b**) Pre-processing form.

As shown in Figure 17, the processing interface offers all the user's pre-processed datasets as processing options, where a metadata summary is displayed for each pre-processed dataset to provide greater detail. This metadata includes the dataset's label, the date it was built, the number of Tweets contained in it after its pre-processing, its size in bytes, the pre-processed fields that the user chose to keep in the dataset, as well as the label from the pre-processed collection dataset and the words used to collect the dataset from Twitter.

When the user decides which pre-processed dataset to process, the platform displays the processing form by pressing the "Select Dataset" button. In this form, the user can execute a word count job on the chosen pre-processed dataset. This task can be completed using the MapReduce or Spark frameworks. The word count job is ready for visualization in the visualization interface after it has been processed. This form also includes a "Suggest" button, which makes the platform recommend the most suitable framework to the user based on the requested processing job, giving a choice between MapReduce and Spark.

Figure 18a depicts the processing form, while Figure 18b depicts the platform's proposal for the chosen dataset ("p-anaemia"). It should be highlighted that the UI always supports the user when it comes to avoiding mistakes such as typing a label that already exists in the collected or generated dataset labels or neglecting to fill out a field. This is accomplished by using user-friendly warnings when the user makes a mistake, also prohibiting the user from submitting the process without a proper processing form. Furthermore, if no pre-processed datasets are available, the interface informs the user that a pre-processing operation is necessary.

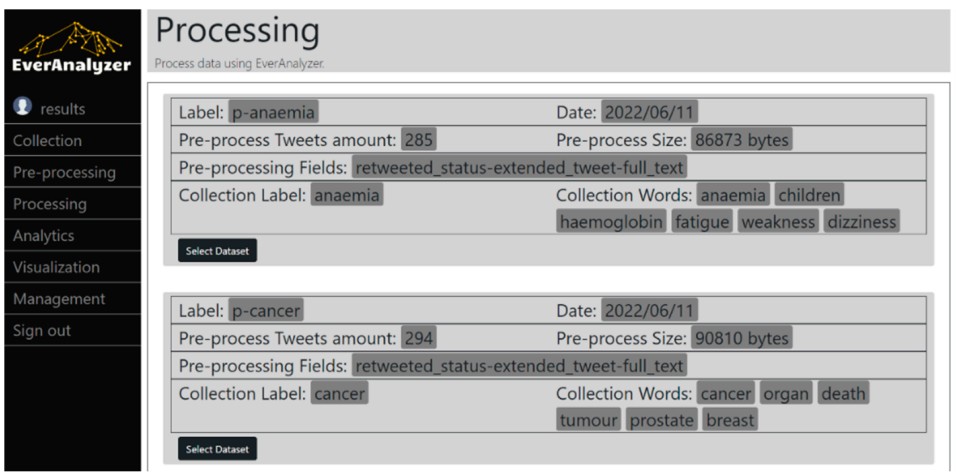

**Figure 17.** Processing Interface—Pre-processing datasets.

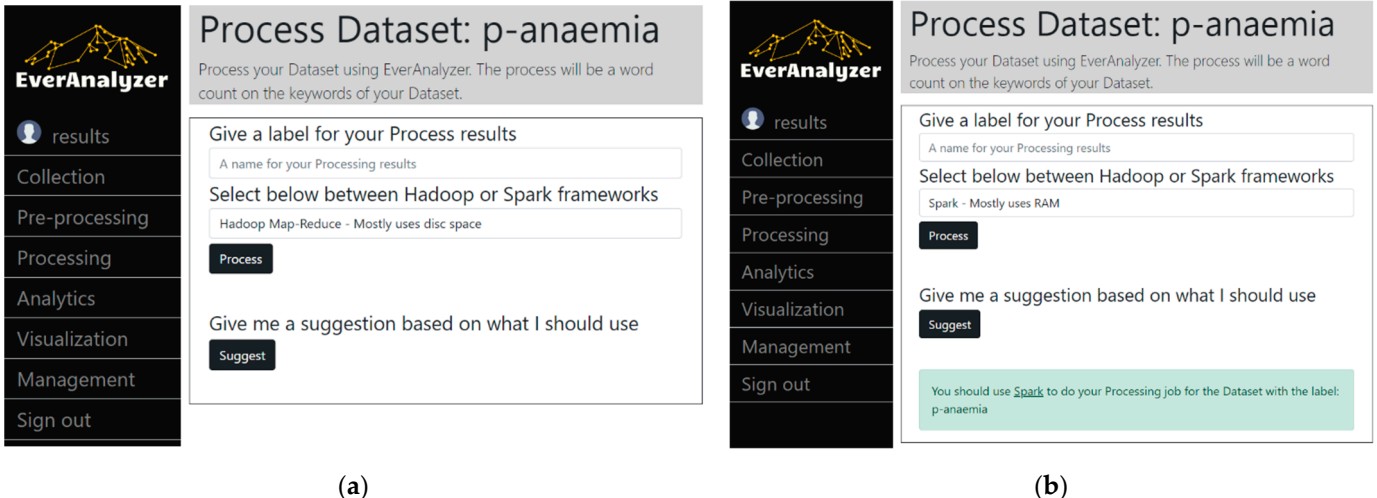

(**a**)          (**b**)

**Figure 18.** (**a**) Processing form; (**b**) Processing proposal.

As shown in Figure 19, the analytics interface then shows all the user's pre-processed datasets as different options for the analytic jobs, where a metadata summary is displayed for each pre-processed dataset. This metadata includes the dataset's label, the date it was built, the number of Tweets contained in it after its pre-processing, its size in bytes, the pre-processed fields that the user chose to keep in the dataset, the label from the pre-processed collection dataset, and the words used to collect the data from Twitter.

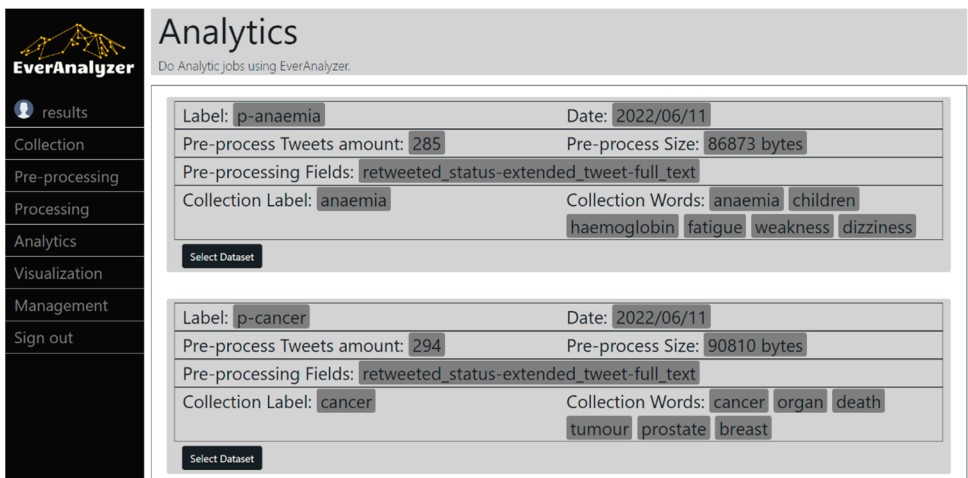

**Figure 19.** Analytics Interface—Pre-processing datasets.

When the user selects a pre-processed dataset to analyze, the platform displays the analysis form by clicking the "Select Dataset" button. In this manner, the user can choose an algorithm to run on the chosen pre-processed dataset, which includes several ML techniques such as the K-means clustering algorithm [83] utilized in the experiment. This task can be done with the help of the Mahout or MLlib libraries. After the analysis, the job is ready for visualizing the exported results in the Visualization Interface. This form also has a "Suggest" button, which causes the platform to offer a framework to the user for the desired analytics job, enabling the user to choose between Mahout and MLlib. Figure 20a shows the analysis form, while Figure 20b shows the platform's proposal for the selected dataset ("p-anaemia"). It should be noted that the UI always assists the user in avoiding mistakes such as typing a label that already exists in the dataset labels or failing to fill out a field. This occurs via the use of user-friendly warnings when the user makes a mistake, e.g., preventing the user from submitting the analysis without a valid analysis form. Moreover, if no pre-processed datasets are available, the interface notifies the user that a pre-processing job is required.

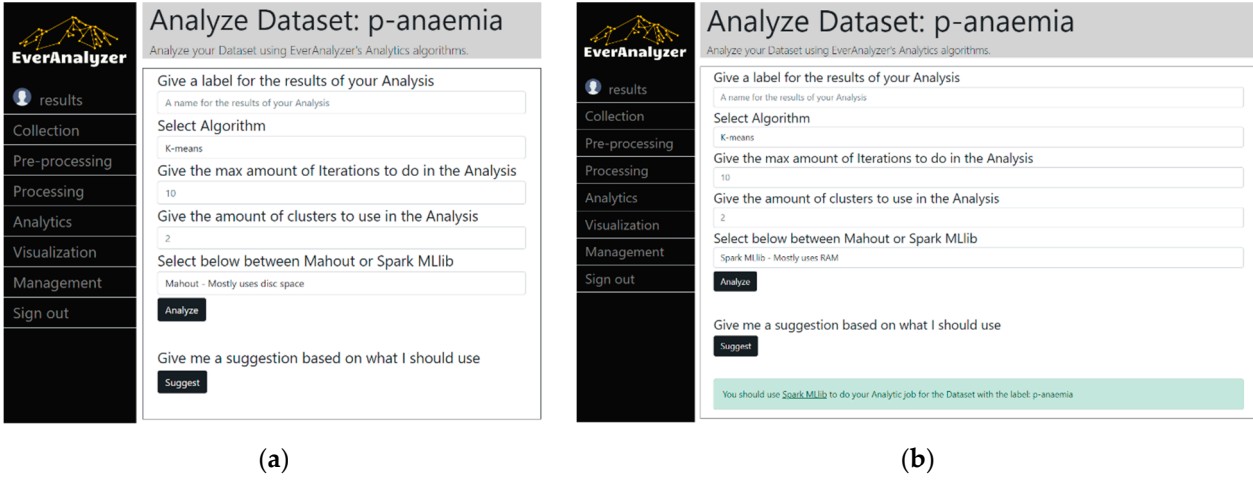

(**a**)  (**b**)

**Figure 20.** (**a**) Analytics form; (**b**) Analytics proposal.

The visualization interface then provides as visualization options all of the user's processed and analyzed results. The datasets are structured in lists that the user can open and dismiss to examine only the processed or analyzed results they are interested in (Figure 21a,b), and the lists warn the user if there are no results to be presented.

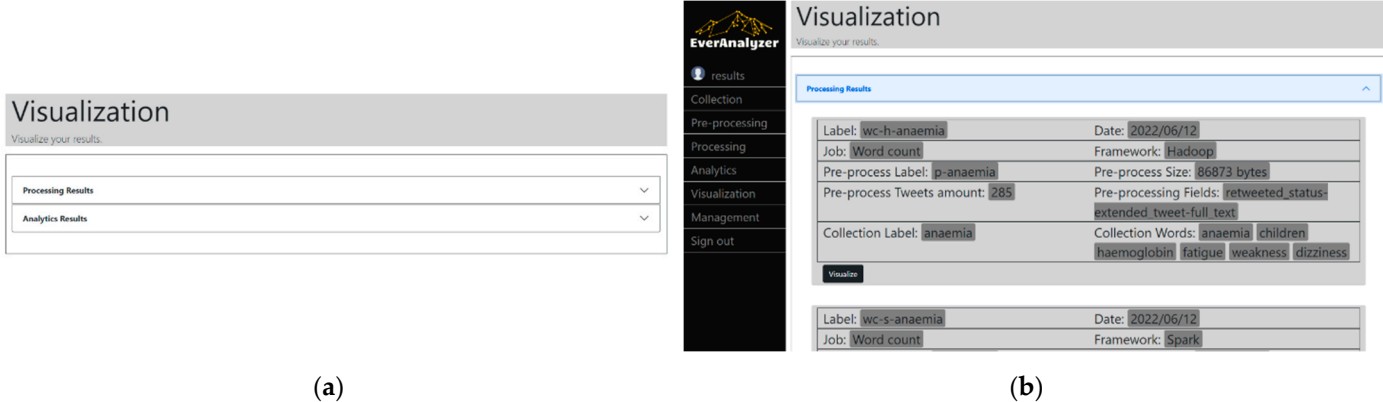

**Figure 21.** (**a**) Visualization lists; (**b**) Visualizable results.

The user can access visualizations of the job that produced the selected result by hitting the "Visualize" button on any of the given outcomes. Because the utility of each visualization method differs, these visualizations differ for each outcome. Figure 22a,b provide two examples of such visualizations, one for the executed processing task and one for an executed analysis task, respectively.

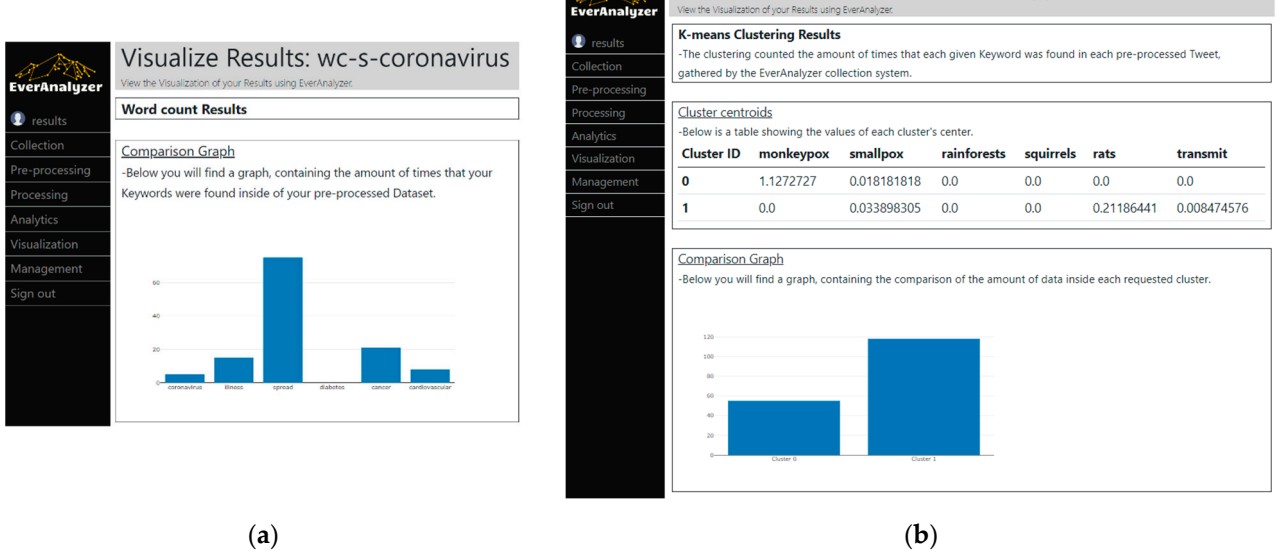

**Figure 22.** (**a**) Visualization of processing task; (**b**) Visualization of analysis task.

In addition, the management interface allows the user to access all of his/her collected, pre-processed, processed, and analyzed datasets and results. It enables the user to view the specified datasets and results as raw data and download all the information from all the completed jobs. An additional functionality is provided for the collected and pre-processed datasets, allowing the user to view all the JSON objects contained within the datasets one by one. As shown in Figure 23a,b, the interface displays all the user's datasets and results as viewing options, organized into lists that the user can open and close to view only one category of datasets, while the lists always notify the user if there are no datasets or results to be displayed. Figure 24a,b illustrates some examples of how the user views EverAnalyzer datasets and results.

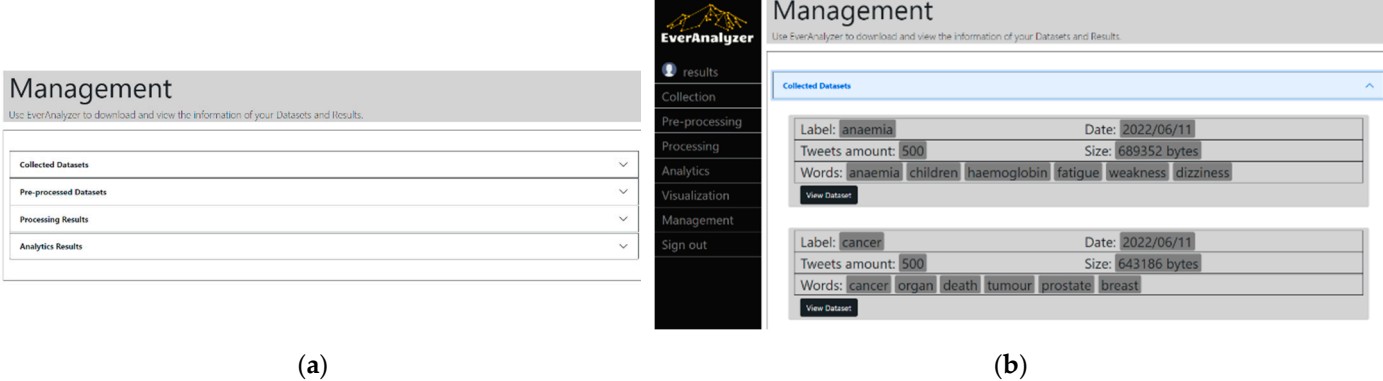

**Figure 23.** (**a**) Management lists; (**b**) Management results.

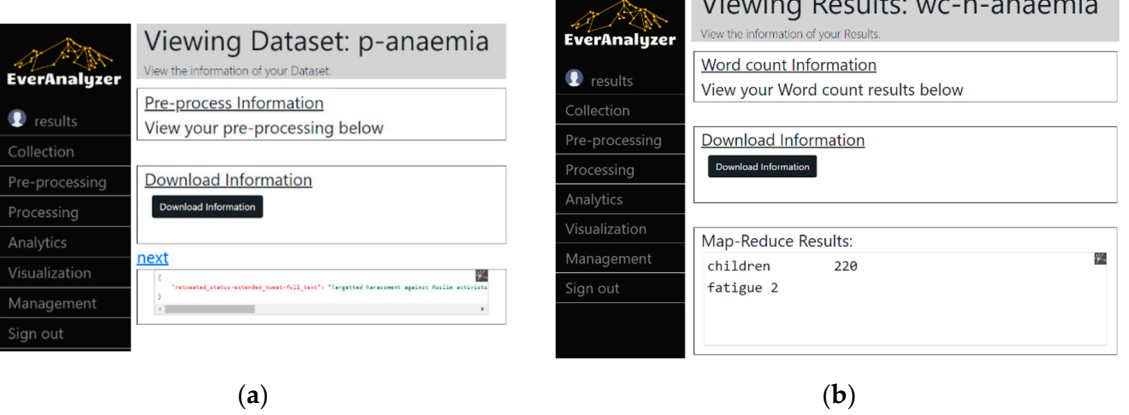

**Figure 24.** (**a**) Viewing pre-processing results; (**b**) Viewing processing results.

### 4.3.2. Performance Evaluation

As mentioned in Section 4.2, the experiment began with the collection of thirty (30) different datasets, each of which contained 500 Tweets based on one of the chosen diseases and conditions. The EverAnalyzer collection interface was used to collect the data, where some words were kept as keywords for each chosen disease and condition to collect the required Tweets. In the Collection Interface, the number of required Tweets, the collected keywords and the user's Twitter Keys and Tokens were provided, as well as a unique label to distinguish the different collected datasets at the end of this process.

Following the collection of the thirty (30) different datasets, EverAnalyzer was used to pre-process all the 15,000 retrieved Tweets, keeping only the fields related to each one's full text. To this end, it should be noted that Twitter records every Tweet as a JSON object, where a JSON object can have many different keys or fields with their respective values in an abstract form. Each value's type can be of any kind, such as int, float, array, or a completely new JSON object. Thus, the pre-processing system in EverAnalyzer detected these JSON objects and found the key path within the JSON object that the user wanted to keep. In the context of the conducted experiment, the retweeted_status/extended_tweet/full_text fields for each retrieved Tweet were kept. This path was specified in EverAnalyzer as "retweeted_status-extended_tweet-full_text", which displayed the path of the keys inside each JSON Tweet by separating it with the minus (-) character. Tweets with no full text were not considered in the next stage of the experiment, because EverAnalyzer detected and marked them as having no value. For each disease and condition, Table 4 shows the total number of Tweets retrieved and saved for further analysis within the EverAnalyzer platform.

**Table 4.** Pre-processing results.

| Disease/ Condition (Twitter Keyword) | Byte Size (Before Pre-Processing) | Byte Size (After Pre-Processing) | Number of Tweets (Before Pre-Processing) | Number of Tweets (After Pre-Processing) |
|---|---|---|---|---|
| Anaemia | 689,352 | 86,873 | 500 | 285 |
| Cancer | 643,186 | 90,810 | 500 | 294 |
| Cholera | 667,851 | 75,305 | 500 | 246 |
| Coronavirus | 682,377 | 67,983 | 500 | 225 |
| Influenza | 635,255 | 80,703 | 500 | 268 |
| Monkeypox | 52,702 | 48,319 | 500 | 173 |
| Obesity | 714,737 | 87,469 | 500 | 285 |
| Pneumonia | 625,063 | 81,848 | 500 | 264 |
| Smallpox | 659,805 | 92,679 | 500 | 299 |
| Syphilis | 158,457 | 86,303 | 500 | 268 |
| Tetanus | 415,683 | 75,669 | 500 | 246 |
| Yellow fever | 82,814 | 60,156 | 500 | 203 |
| Zika virus | 673,913 | 92,458 | 500 | 279 |
| Trachoma | 294,028 | 63,508 | 500 | 205 |
| Diabetes | 659,150 | 49,383 | 500 | 165 |
| Diarrhoea | 679,323 | 96,373 | 500 | 314 |
| Ebola virus | 653,989 | 75,345 | 500 | 241 |
| Epilepsy | 572,757 | 53,907 | 500 | 173 |
| Hepatitis | 612,424 | 84,098 | 500 | 275 |
| HIV-AIDS | 167,690 | 77,874 | 500 | 247 |
| Depression | 721,110 | 70,687 | 500 | 212 |
| Disability | 716,667 | 69,673 | 500 | 218 |
| Cardiovascular | 700,503 | 87,562 | 500 | 294 |
| Chagas | 624,708 | 74,773 | 500 | 251 |
| Dementia | 603,855 | 56,667 | 500 | 185 |
| Dracunculiasis | 119,143 | 78,416 | 500 | 255 |
| Echinococcosis | 617,787 | 67,816 | 500 | 224 |
| Foodborne | 163,814 | 84,306 | 500 | 264 |
| Hypertension | 669,548 | 68,421 | 500 | 224 |
| Infertility | 322,207 | 49,324 | 500 | 155 |

In the next phase, a word count job was performed on each pre-processed dataset's keywords, requesting that the user follow these steps: (i) Request a framework suggestion. (ii) Run the wordcount job with MapReduce. (iii) Run the wordcount job with Spark. To this end, it is important to note that the more datasets EverAnalyzer had, the better its suggestions would be, since it would contain a greater amount of knowledge. Because of steps (ii) and (iii) above, EverAnalyzer was gathering two new datasets as knowledge for each new dataset after the first one. As a result, after pre-processing all the data, the user first asked EverAnalyzer for its recommendation as to which framework (MapReduce or Spark) should be used for the given dataset. Following the platform's response, the user executed two (2) word count jobs for the current dataset. Then, the user wrote EverAnalyzer's proposal and checked if the proposal was the better of the two for each dataset based on execution speed, by first asking EverAnalyzer for the proposition and then performing the execution. Table 5 displays all the captured results.

More specifically, the experiment obtained the following results from Table 5: (i) The number of times EverAnalyzer was able to provide the faster framework for each of the experiments' 30 given datasets. Looking at EverAnalyzer recommendations in Table 5 and the expected recommendations, it is clear that the two columns match in their values on 24 of the 30 rows; every time the rows of the two last columns matched, it meant that EverAnalyzer was able to correctly provide the best suggestion for the user's provided dataset. (ii) The number of times that EverAnalyzer resulted in continuous correct answers over the course of the experiments' 30 datasets. By looking at the result of each suggestion

in Table 5 from top to bottom, it was clear that EverAnalyzer had given one correct answer before giving a wrong one. Then, the platform gave four correct answers before making the next mistake. Table 5 could then show the rest of the times when there were continuous correct answers. (iii) EverAnalyzer's number of correct responses in each consecutive sequence of current replies. Table 5 shows all the occasions when EverAnalyzer made a mistake in the "Suggestion Result" column, and the column "Consecutive Correct Answers" shows the number of correct continuous answers provided by EverAnalyzer. Table 6 describes the execution speed of each data processing operation at the top.

**Table 5.** Processing Results.

| Disease/ Condition | Consecutive Correct Answers | Suggestion Result | EverAnalyzer Recommendation | Expected Recommendation |
|---|---|---|---|---|
| Anaemia | 1 | Correct | MapReduce | MapReduce |
| Cancer | 0 | Wrong | MapReduce | Spark |
| Cholera | 1 | Correct | Spark | Spark |
| Coronavirus | 2 | Correct | Spark | Spark |
| Influenza | 3 | Correct | Spark | Spark |
| Monkeypox | 4 | Correct | Spark | Spark |
| Obesity | 0 | Wrong | MapReduce | Spark |
| Pneumonia | 1 | Correct | Spark | Spark |
| Smallpox | 0 | Wrong | MapReduce | Spark |
| Syphilis | 1 | Correct | Spark | Spark |
| Tetanus | 2 | Correct | Spark | Spark |
| Yellow fever | 3 | Correct | Spark | Spark |
| Zika virus | 0 | Wrong | MapReduce | Spark |
| Trachoma | 1 | Correct | Spark | Spark |
| Diabetes | 2 | Correct | Spark | Spark |
| Diarrhoea | 0 | Wrong | MapReduce | Spark |
| Ebola virus | 1 | Correct | Spark | Spark |
| Epilepsy | 2 | Correct | Spark | Spark |
| Hepatitis | 3 | Correct | Spark | Spark |
| HIV-AIDS | 4 | Correct | Spark | Spark |
| Depression | 5 | Correct | Spark | Spark |
| Disability | 6 | Correct | Spark | Spark |
| Cardiovascular | 0 | Wrong | MapReduce | Spark |
| Chagas | 1 | Correct | Spark | Spark |
| Dementia | 2 | Correct | Spark | Spark |
| Dracunculiasis | 3 | Correct | Spark | Spark |
| Echinococcosis | 4 | Correct | Spark | Spark |
| Foodborne | 5 | Correct | Spark | Spark |
| Hypertension | 6 | Correct | Spark | Spark |
| Infertility | 7 | Correct | Spark | Spark |

An examination of the data and the results provided in Tables 5 and 6 leads to the following conclusions: Firstly, as demonstrated in the literature, Spark performs better with small dataset sizes, with MapReduce not being fast enough to outperform Spark in terms of execution. MapReduce was only able to be faster in the first experiment (Anaemia) due to the time it took for Spark to start its execution. Secondly, EverAnalyzer provided the most efficient proposed framework 24 times out of the 30 times (i.e., 80% success). Of course, it was expected that adding more datasets to the platform would increase this percentage of success. Figure 25 depicts the consecutive correct sequences until a bad proposal was found by the platform. The *x*-axis in the graph represents the number of consecutive correct answers, while the *y*-axis represents the number of consecutive correct suggestions. Although the last pillar has seven consecutive correct answers, it could have had more if additional data had been collected, as no failed suggestions were found in the last streak, and the platform simply ran out of data. As the graph indicates, EverAnalyzer's

continued success appears to be accompanied by an increasing monotony, implying that for more data, its accuracy would be far more than 80%.

**Table 6.** Processing Execution Speeds.

| Disease/Condition | Spark Execution Speed (Milliseconds) | MapReduce Execution Speed (Milliseconds) |
|---|---|---|
| Anaemia | 11,185 | 1387 |
| Cancer | 649 | 1369 |
| Cholera | 469 | 1456 |
| Coronavirus | 474 | 1439 |
| Influenza | 552 | 1414 |
| Monkeypox | 531 | 1358 |
| Obesity | 489 | 1320 |
| Pneumonia | 424 | 1327 |
| Smallpox | 431 | 1416 |
| Syphilis | 427 | 1438 |
| Tetanus | 388 | 1388 |
| Yellow fever | 392 | 1401 |
| Zika virus | 389 | 1300 |
| Trachoma | 354 | 1472 |
| Diabetes | 480 | 1426 |
| Diarrhoea | 374 | 1343 |
| Ebola virus | 496 | 1434 |
| Epilepsy | 434 | 1420 |
| Hepatitis | 325 | 1418 |
| HIV-AIDS | 485 | 1426 |
| Depression | 345 | 1510 |
| Disability | 279 | 1482 |
| Cardiovascular | 291 | 1346 |
| Chagas | 351 | 1469 |
| Dementia | 365 | 1330 |
| Dracunculiasis | 292 | 1430 |
| Echinococcosis | 376 | 1344 |
| Foodborne | 384 | 1441 |
| Hypertension | 382 | 1522 |
| Infertility | 352 | 1317 |

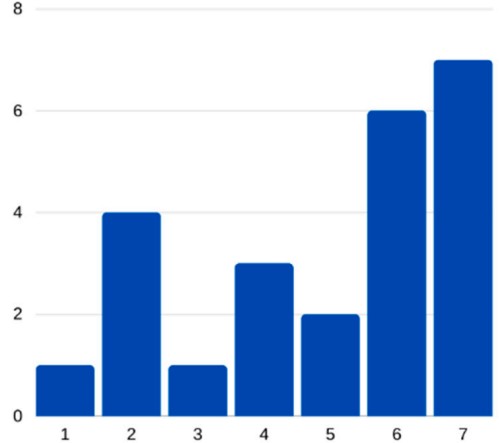

**Figure 25.** EverAnalyzer correct suggestion streaks.

## 5. Discussion

Having thoroughly studied the literature review upon the collection, pre-processing, processing, and analysis of Big Data, it is an undeniable fact that a plethora of research works exist successfully covering such concepts. However, most of them focus on performing the processing and analysis tasks exploiting specific tools and approaches. The following sections depict all the relevant studied literature, concluding with their deficiencies and how EverAnalyzer goes beyond such research, based upon its designed self-adjustable architecture for efficiently processing and analyzing Big Data.

### 5.1. Overall Findings on Big Data Processing

As for the processing tasks, it becomes clear that most of the existing research works make use of either the MapReduce or the Spark frameworks. More specifically, the authors in [84] analyzed Twitter data, focusing on the treatments of MapReduce and Spark of such data, while conducting experimental simulations on the two frameworks. A comparison of two executions in terms of performance and architecture was introduced, along with an analysis to characterize the simulations' conclusions, and the shortcomings and drawbacks of using MapReduce for real-time preparation, demonstrating that Spark was the most useful tool for real-time streaming data. Pirzadeh investigated the time that was spent in various parts of the Hadoop platform in his thesis [85]. Various criteria were simulated and analyzed to understand the platform's behavior while keeping in mind the bottlenecks for effective implementation. Moreover, the authors in [86] conducted log file analysis research on MapReduce and Spark. The authors improved the analysis of applications to realistic log files in both frameworks, and SQL-type queries were performed in real Apache Web Server log files. Furthermore, they conducted various experiments with various parameters to compare and study the act and performance of the two structures and frameworks. The authors in [87] proposed a project based on MapReduce for performing Big Data health analysis, having reported on the useful experience gained from such implementation. What is more, the authors in [88] investigated the theoretical differences and functional comparisons of the Spark and MapReduce platforms. Their findings showed that Spark was much faster for its cache due to duplicate queries like logistic regression. Several Big Data processing techniques were introduced in [89] from system and application perspectives. Cloud data management and Big Data processing mechanisms, such as a cloud computing platform, a cloud architecture, a cloud database, and a data storage scheme, were considered. The authors introduced MapReduce optimization strategies and applications as part of the MapReduce parallel processing framework. In other research [90], the authors attempted to address the issue of detecting anomalies in real-time Big Data processing. They have surveyed the state-of-the-art real-time Big Data processing technologies related to anomaly detection, by first explaining the essential contexts and taxonomy of real-time Big Data processing and anomalous detection, followed by the review of Big Data processing technologies. Finally, they discussed the challenges of real-time Big Data processing in anomaly detection. On the same notion, the study in [91] provided an overview of computing infrastructures for Big Data processing, focusing on architectural, storage, and networking challenges associated with Big Data support. It discussed emerging computing infrastructures and technologies that had the potential to improve data parallelism, task parallelism, and to encourage vertical and horizontal computation parallelism. Research [92] reviewed and discussed the mechanisms for handling and processing Big Data in the healthcare domain, as well as providing a detailed analysis of the mechanisms that are used. It presented a systematic and in-depth examination of cloud computing applications in the healthcare Big Data sector. Also, this research provided implications for research and practice as the direction of future study for healthcare decision-makers. Based on the study's findings, there was sufficient evidence to recommend that cloud computing can provide significant benefits and opportunities to the healthcare sector. The paper [93] investigated a practical case of a Hadoop-based medical Big Data processing system that intelligently processed medical Big Data and uncovered

some characteristics of hospital information system user behaviors. A five-node Hadoop cluster was built to execute distributed MapReduce algorithms, which when compared with single nodes, indicated promise in facilitating efficient data processing with medical Big Data in healthcare services and clinical research.

Rather than choosing one specific tool for performing processing tasks, the study [88] compared the architectures as well as the performances of the MapReduce and the Spark frameworks, concluding with the findings summarized in Table 7.

**Table 7.** MapReduce and Spark differences.

| MapReduce | Spark |
|---|---|
| Inefficient for applications that repeatedly reuse the same set of data. | Uses in-memory processing, reusing it for faster computation. |
| Quite faster in batch processing. | As memory size is limited, it is quite slower in batch processing of huge datasets. |
| Data is stored in disk for processing. | Data stored in main memory. |
| Difficulty in processing and modifying data in real-time due to its high latency. | Processes and modifies data in real-time due to its low latency. |
| Used to process from bygone datasets. | Used for streaming/batch processing and ML. |
| Uses replication for fault tolerance. | Uses Resilient Distributed Datasets (RDDs) for fault tolerance. |
| Merges and partitions shuffle files. | It does not merge and partition shuffle files. |
| Primarily disk-based computation. | Primarily RAM based computation. |

Relevant research was also conducted between the word count and tera-sort processes in nine-node clusters with datasets ranging in size from 600 gigabytes to 600 terabytes [10]. According to the findings, the performance of the MapReduce and Spark systems was largely determined by the sizes of the data entered and the configurations provided to them. Finally, extensive research was conducted in [12] using various HiBench procedures to compare MapReduce and Spark environments, where analytical procedures were carried out within a processing cloud. Spark appeared to outperform MapReduce in almost every experiment, with statistics reaching 92.25% increased efficiency and consuming up to 10% more memory in some cases. Spark did not appear to perform as well with MapReduce in cases where system memory was limited, causing it to make more frequent contact with HDDs. As a result, these findings corroborate those of previous studies that found that Spark stops performing better than MapReduce in large datasets with limited RAM.

*5.2. Overall Findings on Big Data Analysis*

As for the analysis tasks, it becomes clear that most of the existing research works utilize either the Mahout or the MLlib frameworks. To be more specific, the authors in [83] investigated the performance of MapReduce and Spark platforms on ML algorithms. The results of running the K-means algorithm on datasets of various sizes on top of both MapReduce and Spark showed that the runtime of the used algorithm implemented on Spark was 4.5 times faster than that of MapReduce. MapReduce consumed more resources, such as the system's central processor and networking, whereas Spark consumed more RAM than MapReduce. The comparison in [94] focused on various ML frameworks for Big Data, providing a comparison among the ML tools of Mahout, MLlib, H2O, and SAMOA. Furthermore, the authors assessed each tool based on a variety of criteria, including scalability, fault tolerance, and usability. In the research work in [95], a Big Data platform was established by using existing Hadoop ML components such as Mahout and MLlib to realize customer automatic response and information analytic in the field of electric power. It established a Big Data platform that used ML to process massive amounts of electric power customer data, while also developing an automatic response and redirect mechanism prototype. The authors in [96] proposed and implemented a Big Data solution for proactive maintenance in manufacturing. Their architecture consisted of four layers—the data sources, the data transmission, the Big Data analysis, and the visual presentation. For

offline data distribution and calculation, the Hadoop system was used, while Apache Storm was used for real-time processing. The architecture was emphasized in batch and real-time processing, as well as how to use the components for predictive maintenance. In [97], the authors proposed a framework for addressing real-time Big Data management, storage, computation, and predictive data analytics challenges in condition-based maintenance systems to predict and monitor changes in component behavior before they fail. In this regard, Apache Kafka was used as a distributed messaging system to collect unstructured and semi-structured data when dealing with real-time data; the data was collected and delivered to Spark Streaming engine, whilst MLlib was used for data analytics, utilizing HDFS as a file system. The authors in [98] presented a model for dealing with Big Data that relied on a distributed computing environment. Apache Spark as an execution engine and Hive as a database were used in the proposed model. In addition, in their hybrid model, they used HDFS for distributed storage and Spark MLlib for analytic jobs. The implemented model was capable of handling large amounts of data efficiently and could process large datasets and deliver results in real-time.

Research [99] was created by putting the Mahout and MLlib frameworks to the test. According to this study, the range of ML algorithms that the tools support is quite diverse. Table 8 shows the algorithms supported by the frameworks during the reported research.

**Table 8.** Implemented Algorithms of Mahout and MLlib.

| Category | Algorithm | Mahout | MLlib |
|---|---|---|---|
| Dimension Reduction | Principle Component Analysis (PCA) | Yes | Yes |
| | Singular Value Decomposition (SVD) | Yes | Yes |
| Regression | Linear Regression | No | Yes |
| | Logistic Regression | No | Yes |
| Clustering | Hierarchical Clustering | No | Yes |
| | Distributed-based Clustering | No | Yes |
| | Centroid-based Clustering (K-means) | Yes | Yes |
| Classification | Support Vector Machines (SVM) | No | Yes |
| | Artificial Neural Networks (ANN) | No | Yes |
| | Decision Tree | No | Yes |
| | Naive Bayes | Yes | Yes |
| | Ensemble Methods (Boosting, Random Forest) | Yes | Yes |

In the same notion, the MLlib and Mahout frameworks were compared using the K-means, Logistic Regression, and Alternating Least Squares algorithms in the research experiment of [15]. The experiment was carried out by increasing the size of the imported data to a maximum of 10 Gigabytes. The experiments generated these results: (i) MLlib is much faster than Mahout. (ii) MLlib and Mahout become slower by increasing the data. (iii) For MLlib, when data are extremely large, the memory on Spark is not enough to store newly intermediates results, and as a result MLlib crashes. (iv) For Mahout, even if it becomes slow when the data is large, it is always stable. Finally, this study concluded that Mahout was a strict solution if the user wanted to analyze large amounts of data, but MLlib performed better in all the other cases, assuming that it had enough memory to complete its analysis.

### 5.3. Overall Findings of EverAnalyzer

While MapReduce is referred to as a new approach to Big Data processing in modern computing environments, it has also been criticized as a "major step backwards" when compared to DBMSs. As the debate continues, the outcome demonstrates that neither MapReduce nor Spark are particularly satisfactory at what the other does well, and that the two technologies are complementary. However, because of the numerous tools that have been developed to perform common processing tasks, it is critical to select the appropriate tool for the various processing cases. Apart from the comparisons presented above between MapReduce and Spark, on which extensive research has been conducted, there appears not

to have been any attempt to manage data processing using hybrid systems based on the two frameworks. EverAnalyzer tries to fill in this gap by not only providing a solution for a user to manage their data processes using Spark or MapReduce, but also by allowing the platform to advise the users on the most appropriate use of the aforementioned tools. On top of this, according to the literature, there is no reason to experiment with Mahout and MLlib, since it is clear that Mahout should only be used instead of MLlib if the analysis dataset is larger than the RAM of the given system. Because of the simple algorithm (i.e., K-means clustering algorithm) during the experimentation with the EverAnalyzer's proposal system to select the best performance framework for analytic jobs, the results were as expected. Overall, the innovation of EverAnalyzer lies in the fact that the developed system can automatically recognize which one of the underlying data processing (i.e., MapReduce or Spark) and data analysis (i.e., Mahout or MLlib) tools are most suitable and efficient for successfully and more quickly processing and analysis tasks of either batch or streaming ingested data.

Based on all the captured results, EverAnalyzer is expected to perform quite well towards the goal of assisting its users with the management of their Big Data workflows. The platform enables the user to automatically collect, pre-process, process, and analyze data. The results of each processing and analysis job can be visualized, whereas all this information can be downloaded to be used in the future in any way that the user desires. Furthermore, the platform can assist its users by saving them time due to its hybrid nature that makes use of the Apache Hadoop ecosystem, and its suggestion mechanism that is continuously learning with each new processing and analysis job performed by a different user. In an ideal system, the user will always use the most efficient framework for analytic jobs, so the optimal time for the analysis would be the time for the optimal framework of the two to execute the analytic processes on its specified use case. A user who does not know which framework is best for an analysis could try to maximize the chances of selecting the best framework by exploiting the EverAnalyzer platform.

Figure 26a depicts the best (red) and worst (blue) execution times of the two frameworks for each dataset analysis performed, as well as the difference in milliseconds. As a result, the red line represents the ideal time for a user to complete all 30 processes (based on the 30 collected datasets), while the orange line represents the time "saved" from the worst case by performing the subtraction of the worst execution time from the best execution time. It is clear from the graph that the orange line has more than half the area of the blue line if we draw vertical lines from the $x$-axis to the ends of the field that defines it. This means that, from a mathematical standpoint, the user saves more than half of the total execution time by choosing the best framework. Figure 26b, on the other hand, depicts the ideal (red) and worst (blue) analysis times, but it also depicts the time a user would spend if they consistently followed the EverAnalyzer (orange) proposals during the experimentation. This figure combines the data from Figures 25 and 26a by displaying the exact times when EverAnalyzer failed to make a correct suggestion. It is also clear that EverAnalyzer follows the best execution time 80% of the time. Finally, due to EverAnalyzer's good performance (80%) in proposing the fastest framework for each given dataset, it can be seen that EverAnalyzer manages to reduce its users' waiting time for the performed processing jobs. Because EverAnalyzer is data-driven, the more metadata it collects, the closer its execution speeds will be to the ideal red line. However, EverAnalyzer has the disadvantage that if its system exports a wrong suggestion, the user is forced to wait for the worst time to perform the analysis.

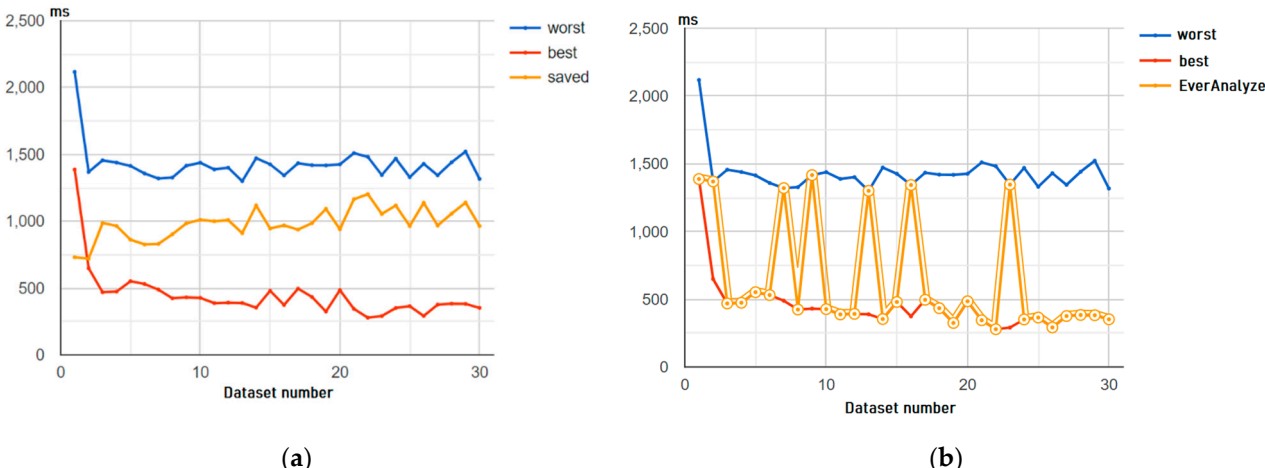

**Figure 26.** (**a**) Worst—best execution with best speed; (**b**) Worst—best execution with EverAnalyzer.

## 6. Conclusions

In this study, a review of the Big Data industry and its tools for management and analysis was conducted, focusing especially on the tools and frameworks that have been developed around the Hadoop ecosystem. A thorough literature review on two of the most popular Big Data processing and analysis tools of this ecosystem has been conducted, referring to MapReduce and Spark, as well as their respective ML libraries, Mahout and MLlib. Despite the comparison performed among such tools, the existing systems that use a combination of the two frameworks for data processing do not appear to have been widely built, so EverAnalyzer is one of the first ones that has tried to implement such an approach. For successfully using such tools in the most suitable scenarios, EverAnalyzer was proposed as a self-adjustable web application for Big Data management. This application aimed to manage users' analysis as fully error-tolerant, preventing them from making mistakes through the use of user-friendly warnings. In addition to fault-tolerant logic, it allowed users to collect, pre-process, process, analyze, and access all of their data. The design of EverAnalyzer tries to provide a better experience for any analyst who wants to perform text mining on text data via Twitter, which is currently one of the most popular and widely used social media platforms for exploiting streaming data.

To verify the applicability and the efficiency of all the above-mentioned functionalities of EverAnalyzer, various experiments were carried out on the platform, aiming to examine the success rates of the platform's suggestion functionality for wordcount jobs. Spark appeared to outperform MapReduce in smaller datasets, as well as datasets smaller than the total RAM of the system that managed them using MLlib, according to the literature. However, MapReduce appeared to outperform in larger datasets, just as Mahout appears to outperform MLlib in datasets larger than the total system RAM used. Finally, Spark appeared to be less fault tolerant than MapReduce, resulting in a more friendly use of MapReduce by those interested in these frameworks, despite the fact that it is not always the best choice in terms of analysis speed.

It is also worth mentioning that the EverAnalyzer platform made use of the existing literature to provide a better experience for its users. There do not appear to be many platforms that use a combination of such frameworks. However, such systems are intended to innovate in the field of Big Data management by providing optimal solutions to data analysts. A representative example of a system that is providing optimal solutions to its users is the Diastema Big Data analytics platform [100,101], which is providing a set of efficient and scalable components that provide user-friendly analytics via graph data modeling and supporting technical and non-technical stakeholders. In the same context, by exploiting Big Data processing and analytics tools and technologies, the PolicyCLOUD data-driven platform exploits added-value of analytics over various datasets to obtain actionable insights and drive decision making [102].

## 6.1. Future Research Directions

In general, it is difficult for a non-IT expert to fully understand every part of Data Science, from developing Big Data software to evaluating and carrying out Big Data studies. EverAnalyzer is a solution that tries to contribute to the aforementioned issue, relieving the strain on data analysts, data scientists, and data engineers, facilitating at the same time the work of all non-IT users. This platform is an infant step in comparison to the tremendous growth of the Big Data management sector. Thus, building Big Data management platforms like EverAnalyzer and attempting to integrate various software and functions on them would be quite valuable for future research. Such functions could include the option to rename the fields inside the JSON documents of the collected and created datasets, also enabling addition of a field with some specific values in a user's datasets. In addition, a more dynamic Visualization System would be highly beneficial for data interpretation by users, whereas additional valuable enhancements could include the capacity of the platform to collect data from more social media networks, such as Facebook, Instagram, Tik Tok, and Bereal, as well as from other sources that expose open data to the external world. Besides, testing such platforms in distributed environments would be of great research importance, since Spark and Hadoop frameworks, as well as their accompanying libraries, are designed for such scenarios. As a result, systems such as EverAnalyzer can behave substantially differently when it comes to the execution speed for the requested processing and analytical processes. The same notion goes for the cloud environments as well, where there is great research value in conducting relevant experiments and obtaining results. Finally, experimenting with health data and the capabilities of the framework proposed for processing EverAnalyzer jobs looks to have yielded promising results, and thus it would be very useful to perform further experiments exploiting a larger number of datasets towards achieving better healthcare decision-making.

## 6.2. Research Limitations

This document introduces EverAnalyzer, which makes use of a hybrid system that exploits Spark and MapReduce, leaving, however, a variety of areas that could be improved. In this context, the most significant limitation is the system's limited computing resources. As previously stated, the EverAnalyzer platform was tested on a computer with 237 GB of HDD and eight GB of RAM. When referring to Big Data, these sizes are very small, as seen by the data sizes mentioned in the research's introduction. Additionally, despite the fact that the sample size of 30 distinct datasets is statistically significant, there may be values in datasets that are handled differently by the two frameworks. Experimenting with this number of datasets is considered minor compared to the enormous number of distinct datasets available from a variety of sources. Another research limitation could be that the current research has been experimented on and evaluated considering mainly the healthcare domain. Despite the fact that the overall system will equally perform in other scenarios and domains according to the extracted insights, additional domain-agnostic experiments, functional evaluations, and performance testing could make this assumption clearer, and sufficiently widen the applicability of EverAnalyzer.

**Author Contributions:** Conceptualization, P.K. and A.M.; methodology P.K. and A.M.; software, P.K. and A.M.; validation, A.M.; formal analysis, A.M., A.K. and D.K.; investigation, A.M. and A.K.; resources, P.K.; data curation, A.K.; writing—original draft preparation, P.K., A.M. and A.K.; writing—review and editing, P.K., A.M., A.K. and D.K.; visualization, A.K.; supervision, D.K.; project administration, D.K.; funding acquisition, D.K. All authors have read and agreed to the published version of the manuscript.

**Funding:** The research leading to the results presented in this paper has received funding from the European Union's funded Project iHELP under grant agreement no 101017441, and from Greek national funds through the Operational Program on Competitiveness, Entrepreneurship and Innovation, under the call RESEARCH—CREATE—INNOVATE: T2EDK-04612.

**Institutional Review Board Statement:** Not applicable.

**Informed Consent Statement:** Not applicable.

**Data Availability Statement:** Not applicable.

**Conflicts of Interest:** The authors declare no conflict of interest.

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
