# Peer review of "EverAnalyzer: A Self-Adjustable Big Data Management Platform Exploiting the Hadoop Ecosystem"

_information, doi:10.3390/info14020093_

Round 1

Reviewer 1 Report

this paper illustrates developing a platform that has integrated several tools with both Spark and MapReduce. While explained in details and sounds, the text needs a native speaker to do a thoughtful editing for a better quality.

Author Response

Please see the attached pdf file

Reviewer 2 Report

I see some merit in the actual contents of the paper. However, I think Author should improve the organization and the contents of the paper.

1.              Introduction. Introduction should show paper motivation, paper purpose and which is the paper knowledge contribution. After reading it, the research objectives and their importance are hidden. Paper motivations could be considerably strengthened by providing evidence, in practice and in theory, as to why is necessary to develop this proposal. The paper, as it currently stands, isn’t strongly motivated in terms of how it meets an existing gap in the literature. 

Some research questions or paper’s aims should be included in the introduction. It will help readers to understand the objective of your work. Otherwise, paper seems a technical report of the features of the Big Data management platform developed (EverAnalyzer)

2.     Literature review. Section 2 Material and methods should be renamed as Literature review. Only sections 2.1 – 2.4 should be here

3.     A new section 3 called  Proposed Big Data Management Platform is needed. In its current form, authors are mixing the literature review with their proposal.

4.     Section 3 should be renamed as section 4 Case example

5.     Conclusion section. "Future research directions" and "Research limitations" are crucial for the quality of the research. They are not stated in this version, please add them at the end of the paper.

Author Response

Please see the attached pdf file

Round 2

Reviewer 2 Report

In my opinion, paper is now ready for publication

Author Response

We would like to thank the reviewer for the overall effort and the valuable comments we received, in order to improve the quality of our research work.